# Migrating diurnal tide anomalies during QBO disruptions in 2016 and 2020: morphology and mechanism

Shuai Liu[1,2], Guoying Jiang[1,3,4], Bingxian Luo[1,2], Xiao Liu[5], Jiyao Xu[1,3], Yajun Zhu[1,3,4], Wen Yi[6,7]

[1]State Key Laboratory of Solar Activity and Space Weather, National Space Science Center, Chinese Academy of Sciences, Beijing, 100190, China
[2]College of Earth and Planetary Sciences, University of Chinese Academy of Sciences, Beijing, 101408, China
[3]School of Astronomy and Space Science, University of Chinese Academy of Sciences, Beijing, 101408, China
[4]Hainan National Field Science Observation and Research Observatory for Space Weather, Danzhou, Hainan Province, China.
[5]School of Mathematics and Statistics, Henan Normal University, Xinxiang, 453007, China
[6]CAS Key Laboratory of Geospace Environment, Department of Geophysics and Planetary Sciences, University of Science and Technology of China, Hefei, China
[7]CAS Center for Excellence in Comparative Planetology, Anhui Mengcheng Geophysics National Observation and Research Station, University of Science and Technology of China, Hefei, China

*Correspondence to*: Guoying Jiang (gyjiang@swl.ac.cn) and Bingxian Luo (luobx@nssc.ac.cn)

**Abstract.** The stratosphere Quasi-Biennial Oscillation (QBO) modulates the migrating diurnal tide (DW1) in the mesosphere and lower thermosphere (MLT). DW1 amplitudes are larger during QBO westerly (QBOW) than during easterly (QBOE) phases. Since QBO's discovery in 1953, two rare QBO disruption events occurred in 2016 and 2020. During these events, anomalous westerly winds propagate upward, disrupting normal downward propagation of easterly phase and producing a persistent westerly wind layer. In this study, global responses of DW1 amplitudes and phases in MLT to these QBO disruptions, as well as their underlying mechanisms are investigated, using SABER/TIMED observations, MERRA-2 reanalysis and SD-WACCM-X simulations. Similarity of the DW1 responses to these two events is that DW1 phases and wavelengths exhibit close results to QBOW, whereas amplitudes lie between QBOW and QBOE. DW1 amplitudes in equatorial MLT (maximum difference) vary by 20.5% and -10.2% relative to QBOE and QBOW in 2016 event, but by 6.0% and -21.1% in 2020 event. In 2016 event, water-vapor radiative and latent heating, jointly modulated by the event and ENSO, increase significantly relative to QBOE. Less dissipation and less tidal energy removal in stratosphere along with gravity-wave (GW) drag in mesosphere tend to enhance DW1 amplitudes. In contrast, in 2020 event, water-vapor heating exhibits a 5% increase. The dissipation and tidal energy removal has adverse effect on DW1, while GW drag exerts a weaker influence. The enhancement of water vapor heating together with the weaker GW drag likely accounts for the weaker enhancement of DW1 during this event.

## 1 Introduction

Atmospheric solar tides are planetary-scale harmonic waves with periods of a solar day. In the mesosphere and lower thermosphere (MLT), solar tides exert significant influences on atmospheric parameters such as wind, temperature, and density (Chapman & Lindzen, 1980; Xu et al., 2009; Jiang et al., 2010; Smith, 2012). Among these tides, the migrating diurnal tide (DW1) is one of the most prominent components. DW1 in MLT is modulated by external forcings, including the stratosphere Quasi-Biennial Oscillation (QBO, Hagan et al., 1999; Wu et al., 2008; Xu et al., 2009; Oberheide et al., 2009; Mukhtarov et al., 2009; Davis et al., 2013; Gan et al. 2014), El Niño–Southern Oscillation (ENSO, Lieberman et al., 2007; Cen et al., 2022) and 11-year solar cycle response (Singh and Gurubaran, 2017; Sun et al., 2022; Liu et al., 2024a; Liu et al., 2024b). In this work, the impact of QBO is focused on.

The QBO dominates the variability of the equatorial stratosphere (∼16–50 km), shown as alternating downward propagating easterly wind (so-called QBO easterly phases) and westerly wind (so-called QBO westerly phases), with an averaging period of approximately 28 months (Baldwin et al., 2001). QBO is driven by vertically propagating Kelvin waves, mixed Rossby gravity waves and small-scale gravity waves (Lindzen and Holton, 1968; Holton and Lindzen, 1972; Baldwin et al., 2001; Ern et al., 2014). It could influence the transport and distribution of trace gases like water vapor and ozone in the troposphere and stratosphere (Schoeberl et al., 2008).

During the winter of 2015/16 and 2019/20, two rare stratospheric QBO disruption events occurred, which were found only twice since the record began in 1953. The events are manifested by anomalous westerly winds propagating upward, disrupting normal downward propagation of the easterly phase and producing a persistent westerly wind layer (Newman et al., 2016; Anstey et al., 2021). The 2016 QBO disruption has been confirmed to have a close causal relationship with the 2015/16 extreme El Niño event (Newman et al., 2016; Osprey et al., 2016; Barton and McCormack, 2017; Coy et al., 2017). The 2015/16 El Niño substantially weakened the subtropical easterly jet, allowing enhanced Rossby wave propagation from the extratropics into the deep tropics near 40 hPa (Barton and McCormack, 2017). These amplified Rossby waves subsequently broke and deposited momentum near the QBO westerly core, rather than at the climatological zero-wind line, causing a pronounced deceleration. The deceleration gave rise to a persistence of westerlies at 40–15 hPa, preventing the expected transition to easterlies and ultimately leading to the QBO disruption (Newman et al., 2016; Osprey et al., 2016; Coy et al., 2017; Barton and McCormack, 2017; Kang et al., 2022; Wang et al., 2023). The QBO disruption was accompanied by a marked strengthening of the Brewer–Dobson residual circulation, thereby intensifying tropical upwelling. This upwelling contributed to an upward displacement of westerlies in the tropical lower stratosphere (Coy et al., 2017), modifying the transport and distribution of trace gases such as water vapor. The persistent westerlies also created conducive background conditions for the vertical propagation of DW1. Nevertheless, not all strong El Niño events trigger QBO disruptions. In the 2015/16 case, the QBO westerly wind core was weaker and Rossby wave activity was stronger than in other extreme events, such as the 1998 El Niño (Barton and McCormack, 2017). In the 2020 event, the upward-propagating westerly wind was so weak that the monthly mean zonal wind appeared as upward-propagating easterly wind (e.g., Anstey et al., 2021; Wang et al., 2023). This

event was driven by strong extratropical Rossby waves associated with the 2019 minor SSW in the southern hemisphere (Kang
and Chun, 2021; Wang et al., 2023). In these two events, the trace gases such as ozone and water vapor are modulated. During
the 2016 QBO disruption event, positive water vapor anomalies were observed between the tropopause and lower stratosphere,
while positive ozone anomalies appeared in the upper stratosphere (Tweedy et al., 2017; Diallo et al., 2018). A similar pattern
was reported for the 2020 disruption event, with water vapor in the lower stratosphere and ozone in the upper stratosphere also
exhibiting positive anomalies (Diallo et al., 2022).
QBO modulation of diurnal tides has been reported by both ground-based and space-borne observations (Araújo et al., 2017;
Davis et al., 2013; Pramitha et al., 2021b; Wu et al., 2008; Dhadly et al., 2018). Mayr and Mengel (2005) reported that the
QBO can affect these amplitudes by up to 30% using the Numerical Spectral Model (NSM). Thermosphere, Ionosphere,
Mesosphere Energetics and Dynamics/Sounding of the Atmosphere using Broadband Emission Radiometry (TIMED/SABER)
observations revealed that the quasi-biennial variability of DW1 could exceed 50% at certain altitudes (Garcia, 2023). The
modulation was characterized by larger-than-average diurnal tide amplitudes during the westerly phase of the QBO and
smaller-than-average amplitudes during the easterly phase (Vincent et al., 1998; Wu et al., 2008; Xu et al., 2009; Davis et al.,
2013; Araújo et al., 2017; Pramitha et al., 2021b; Garcia, 2023). Several mechanisms have been proposed for modulating the
migrating diurnal tide (DW1). A primary factor emphasized in many studies is the variation in the background zonal wind and
its latitudinal shear (Forbes and Vincent, 1989; Hagan et al., 1999; McLandress, 2002b; Riggin and Lieberman, 2013; Liu et
al., 2015; Ortland, 2017; Dhadly et al., 2018; Pramitha et al., 2021a, b). Forbes and Vincent (1989) demonstrated that the DW1
(1,1) mode experiences stronger dissipation in easterly phases than in westerly phases, while McLandress (2002b) highlighted
the tide's strong sensitivity to latitudinal shears in the zonal mean easterlies of the summer mesosphere. Apart from the
influence of the background wind, additional contributions have been suggested, including variations in diurnal heating
(McLandress, 2002b; Riggin and Lieberman, 2013; Ortland, 2017) and tide–gravity wave (GW) interactions (Mayr et al., 1998;
McLandress, 2002a; Lu et al., 2012; Wang et al., 2024), both of which may play a role in modulating the QBO-related
variability of DW1.
Recent studies have shown that the diurnal tides were modulated during the QBO disruption events (Pramitha et al., 2021a;
Garcia, 2023; Wang et al., 2024). Pramitha et al. (2021a) first reported the enhancement of the diurnal tides during the
2015/2016 QBO disruption event using a meteor radar over Tirupati (13.63°N, 79.4°E) and linked this enhancement to changes
in ozone concentration. Garcia (2023) showed the equatorial response of temperature DW1 to these two disruption events
when analysing the QBO modulation to DW1. Wang et al. (2024) reported the weakened mesospheric diurnal tides at mid-
latitude during QBO disruption events, which were observed by a meteor radar chain. They further provided the modulation
evidence of gravity wave forcing and solar radiative absorption by subtropical stratospheric ozone, as revealed by SD-
WACCM-X simulations.
These findings raise three questions: (1) In addition to the equatorial peak, temperature DW1 exhibits secondary amplitude
maxima at 30°N and 30°S (Xu et al., 2009; Garcia, 2023). Whether the DW1 amplitudes on a global scale show a similar
response to the QBO disruption events. (2) Whether the phases and wavelengths of DW1 could be affected by the events. (3)
Mechanisms for modulating DW1 include heating sources such as water vapor radiative heating and latent heating, zonal wind
latitudinal shear, and tide–gravity wave interactions (e.g., Forbes and Vincent, 1989; Hagan, 1996; Hagan et al., 1999;
McLandress, 2002a; Kogure and Liu, 2021). Whether these mechanisms play significant roles in modulating DW1 during
QBO disruption events.
The present study will focus on the global response feature of DW1 and its underlying mechanisms to QBO disruption events.
The response of DW1 amplitudes, phases and wavelengths during the event will be investigated. Moreover, the contribution
of possible mechanisms, including heating sources, the zonal wind latitudinal shear and tidal-gravity wave interaction during
the event, will be explored. The article is organized as follows: Section 2 introduces TIMED/SABER, SD-WACCM-X, and
MERRA-2 data and the methodologies to extract the migrating tides. Section 3 presents the response feature of the DW1 to
the QBO disruption events revealed by SABER/TIMED observations and SD-WACCM-X simulation results. The possible
mechanism of DW1 response to the disruption events is discussed in Section 4. Section 5 presents the summary.
**2 Data and methodology**
This study employs the dataset of SABER/TIMED observations, SD-WACCM-X simulations and MERRA-2 reanalysis to
reveal the feature of DW1 and its excitation sources during QBO disruption events. DW1 amplitude, phase, and wavelength
are derived from both SABER/TIMED data and SD-WACCM-X outputs. MERRA-2 reanalysis is used to analyse the
contributions of water vapor radiative heating and latent heating to DW1 variability during the QBO disruption events, while
SABER/TIMED observations characterize ozone radiative heating. SD-WACCM-X simulations validate the excitation source
revealed by the observational datasets.
**2.1 SABER/TIMED observations**
The TIMED satellite is in a near sun-synchronous orbit with a 73° inclination at about 625 km. The number of orbits observed
per day is about 15. SABER, an instrument in the TIMED satellite, is a 10-channel broadband (1.27–17 μm) limb-scanning
infrared radiometer. SABER observations of infrared radiance are used to retrieve kinetic temperature, trace gases, etc. In this
work, kinetic temperature and ozone observations in level 2 A (L2A) dataset and ozone heating rate in level 2B (L2B) dataset
are selected to analyse the DW1 response to QBO disruption events. Kinetic temperature is derived using a full nonlocal
thermodynamic equilibrium (non-LTE) inversion algorithm (Mertens et al., 2001; 2004) with the combination of the measured
15 μm $CO_2$ vertical emission profile and $CO_2$ concentrations provided by the Whole Atmosphere Community Climate Model
(WACCM 3.5.48), as described by Garcia et al. (2007).
It takes SABER 60 days to sample 24 hours in local time. The data latitudinal coverage every 60 days extends from 53°N to
83°S or 53°S–83°N. Temperature observations taken from version 2.07 data from 2002 to 2019 and version 2.08 data from
2020 to 2023 are used. The details of the version switches could refer to Mlynczak et al. (2022, 2003). The retrieved
temperature observations used in this work cover altitudes from approximately 15 km to 105 km.

## 2.2 SD-WACCM-X

The Whole Atmosphere Community Climate Model with thermosphere–ionosphere eXtension (WACCM-X) is a comprehensive numerical model that could simulate the Earth's atmosphere from the surface up to the upper thermosphere (~500–700 km), including the ionosphere (Liu et al., 2010; 2018). WACCM-X is a single, unified whole-atmosphere model that extends the NCAR Whole Atmosphere Community Climate Model (WACCM4; Marsh et al., 2013). WACCM4 itself was built upon the Community Atmosphere Model 4 (CAM4; Neale et al., 2013). While the thermosphere–ionosphere physics (e.g., global electrodynamo, $O^+$ transport, electron/ion energetics) incorporated in WACCM-X were largely adapted from the NCAR Thermosphere–Ionosphere–Electrodynamics General Circulation Model (TIE-GCM; Qian et al., 2014; Pedatella, 2022), they have been re-engineered within the WACCM-X dynamical core and coupled to the lower- and middle-atmosphere processes through a dedicated ionosphere-interface module. SD in the SD-WACCM-X means specified dynamics, which is an approach described in Smith et al. (2017). The reanalysis fields from Modern-Era Retrospective analysis for Research and Applications, Version 2 (MERRA-2, Gelaro et al., 2017) data from the surface up to ~50 km are nudged in WACCM-X. Model parameters are output in 3-hour resolution. The latitude-longitude resolution is 1.9°×2.5°. The model has 145 pressure levels with a varying vertical resolution of ~1.1–1.75 km in the troposphere and stratosphere and ~3.5 km in the mesosphere. In this work, the temperature, zonal wind, temperature tendency due to moist process and long wave heating rate ranging from 2002 to 2022 are selected.

## 2.3 MERRA-2

MERRA-2 is a reanalysis product from the NASA Global Modeling and Assimilation Office (GMAO) and provides data like wind, temperature, mixing ratio of components, and so on. (Gelaro et al., 2017). In this work, the zonal wind, temperature, air density, surface albedo, water vapor mixing ratio and temperature tendency due to moist process range from 2002 to 2023 are selected. The time resolution is 3-hour per day. The spatial resolution is a 2.5°×2.5° latitude-by-longitude grid at 72 model levels from ground to 0.01 hPa.

## 2.4 Singapore radiosonde QBO index

The QBO index employed in this study is derived from Singapore radiosonde measurements obtained by the Meteorological Service Singapore Upper Air Observatory (station 48698; 1.34°N, 103.89°E; 21 m above mean sea level). The monthly mean zonal wind data processed by the National Aeronautics and Space Administration/Goddard Space Flight Center (NASA/GSFC) is selected, spanning 2002–2023 at pressure levels between 100 hPa and 10 hPa.

## 2.5 Water vapor radiative heating rate calculation

Troposphere heating by water vapor absorption of near-infrared radiation is an important excitation source for DW1 (Hagan, 1996; Lieberman et al., 2003). Due to the SABER's observational gap in the troposphere, the MERRA-2 dataset is adopted.

In this dataset, temperature, air density, surface albedo, cloud fraction and water vapor mixing ratio (specific humidity) are the
variables necessary for the calculation. The heating rate is the sum of clear sky and cloudy sky (Groves et al., 1982):
$$J = (1 - k)J_{clear} + kJ_{cloudy} \qquad (1)$$

where k is the cloud fraction, $J_{clear}$ and $J_{cloudy}$ are the heating rates of the clear sky and cloudy sky. The calculation equations
for clear sky and cloudy sky are given in Appendix A.

## 2.6 Ozone radiative heating rate calculation

The calculation of ozone radiative heating follows the Strobel/Zhu scheme (Strobel, 1978; Zhu, 1994), in which the total
heating rate is obtained as the sum of contributions from the Hartley, Huggins, and Chappuis bands, with parameterizations
from Zhu (1994). The required ozone volume mixing ratio (VMR) and density are taken from the SABER L2A dataset. Ozone
VMR is retrieved from vertical emission profiles at 9.6 μm and 1.27 μm (Smith et al., 2013). The former covers all local times
and the latter is limited to daytime. In this study, the 9.6 μm retrievals are used. It should be noted that the Strobel/Zhu model
omits the dominant nighttime chemical-heating source between ~70 and 100 km (Zhu, 1994; Xu et al., 2010). Consequently,
the present analysis is restricted to the sum of the three-band heating rates between 20 and 70 km.

## 2.7 Method for extracting DW1 and data processing

Non-uniform SABER observational data were processed into zonal mean data and used to extract tides. The procedures are
briefly introduced as follows. Firstly, the kinetic temperature, ozone mixing ratio and ozone radiative heating rate profiles are
interpolated vertically with a 1 km spacing. Profiles for each day are sorted into ascending and descending groups. Secondly,
the global temperature and ozone observations at whole heights and in both groups were processed into zonal mean results,
covering latitudes from 50°S to 50°N with a resolution of 5°. At a fixed latitude and height, the following equation proposed
by Xu et al. (2007) is used to extract the tide from the zonal mean temperature in a 60-day window:
$$\frac{1}{2\pi}\int_0^{2\pi} T(t_{LT}, \lambda)d\lambda = \bar{T} + \eta(t - t_0) + \sum_{n=1}^{N} A_n cos(n\omega t_{LT}) + \sum_{n=1}^{N} B_n sin(n\omega t_{LT}) \qquad (2)$$

where $\omega = 2\pi/24$(hour), $t_{LT}$ is the local time, $\lambda$ is longitude in radians. $\bar{T}$ is the 60-day window average of the zonal mean
temperature. $\eta$ describes the linear trend variation in the window. t is the day of the window and $t_0$ is the center day of
the window. The third and fourth terms of the right section of the equation denotes the superimposed harmonic signals by four
periods migrating tides, including diurnal tide (DW1), semidiurnal tide (SW2), terdiurnal tide (TW3), and 6-h tide (QW4). N
in the third term represents four signals and n denotes each signal. The amplitude and phase of each migrating tide are retrieved
using $\sqrt{A_n^2 + B_n^2}$ and $arctan(B_n/A_n)$, respectively. The overlapping analyses are obtained by sliding the 60-day window
forward in 1-day intervals to obtain the daily values of the wave characteristics. The details of the methods used for data
processing and tide extraction could refer to Xu et al. (2007, 2009) and Liu et al. (2024a).
The method for extracting tidal components from ozone heating rates follows Equation 4 in Xu et al. (2010). The methods for
tidal extraction from MERRA-2 and SD-WACCM-X differ from those used for SABER due to differences in data structure.
Unlike SABER, both MERRA-2 and SD-WACCM-X provide spatially uniform data with a 3-hour temporal resolution. As a
result, a two-dimensional Fast Fourier Transform (2D-FFT) is directly applied to extract daily DW1 amplitudes and phases of
temperature, water vapor heating rate, and temperature tendency due to moist processes. For further analysis, the Hough mode
decomposition is applied to the DW1. The program is retrieved from https://github.com/masaru-kogure/Hough_Function. As
in Sakazaki (2013), DW1 in the stratosphere can be reasonably well represented by a superposition of only a few ($\sim 4$) Hough
modes. Here the (1, -2), (1, -1), (1, 1) and (1, 2) modes are used. The monthly mean temperature DW1 amplitudes obtained
from SABER, MERRA-2 and SD-WACCM-X are calculated. Due to the observational gap of SABER, the Generalized Lomb-
Scargle Periodogram (from PyAstronomy) is applied to fill the missing data of ozone heating rate. A low-pass Butterworth
filter of 3rd order with a cut-off period of 13 months ($\approx 0.077$ cycles month$^{-1}$) is applied to reveal the DW1 QBO variations
(temperature, ozone heating and so on).

## 3 Result

### 3.1 DW1 amplitude response to QBO disruption events

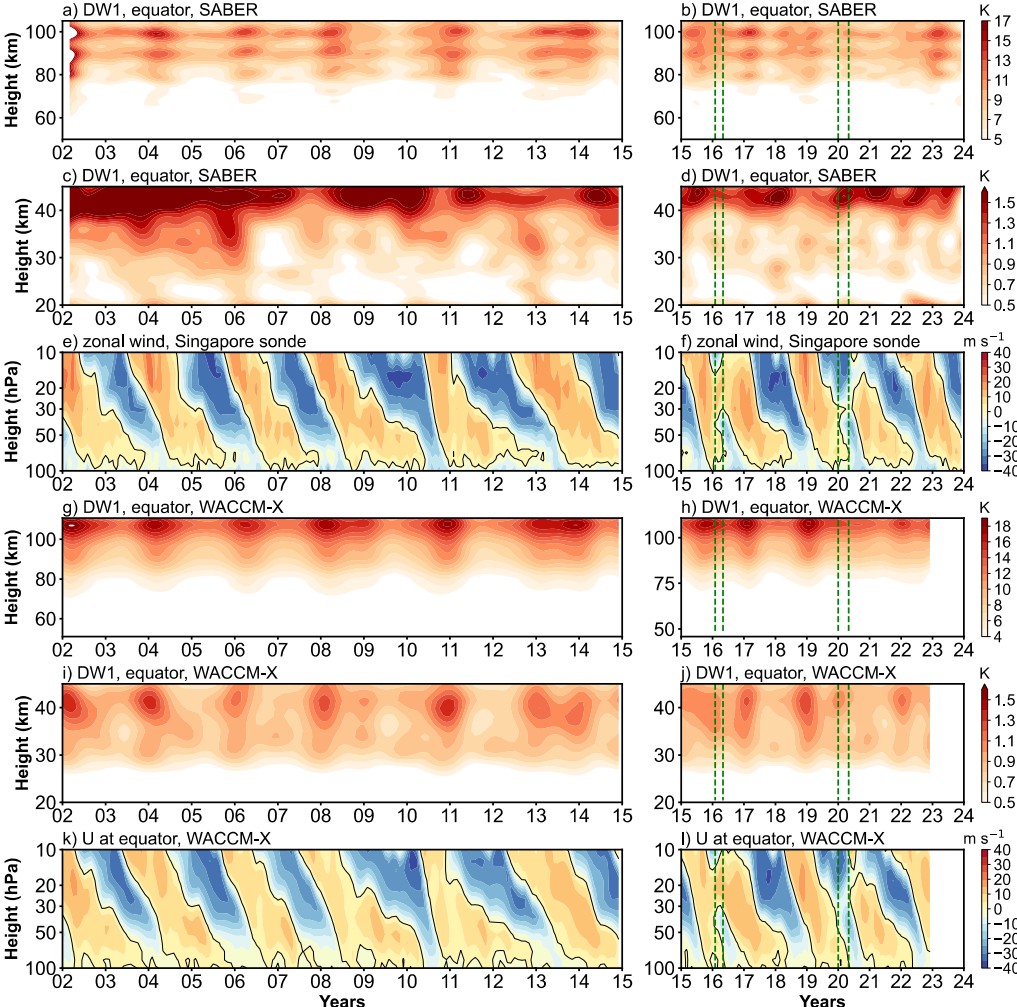

**Figure 1. (a, b)** Low-pass filtered amplitudes (periods longer than 13 months) of the migrating diurnal tide (DW1; monthly mean, in K) as a function of altitude in the mesosphere and lower thermosphere (MLT) and time (2002–2023), derived from SABER/TIMED temperature observations. **(c, d)** Same as (a, b) but for the stratosphere. **(e, f)** Zonal wind at the stratospheric equator from Singapore sonde. **(g–i)** Similar to (a–f), but based on SD-WACCM-X simulations. Vertical green dashed lines indicate the QBO disruption periods in 2015/16 (February–May 2016) and 2019/20 (January–May 2020).

Figure 1 presents the amplitude of DW1 after low-pass filtering and the zonal wind observed by the Singapore sonde. Only amplitude components with periods longer than 13 months are retained. In the stratosphere, the zonal wind shows alternating downward propagating westerly wind (positive value in Figure 1e and 1f) and easterly wind (negative value in Figure 1e and

1f). Each westerly and easterly transition can be called a QBO cycle. In the stratosphere (Figure 1c and 1d), below 40 km, the
amplitude of DW1 also shows Quasi-Biennial variability. Above 40 km, the variation becomes more complex. This feature
will be discussed later. In the MLT region (Figure 1a and 1b), the low-pass filtering results of DW1 at the equator exhibit
Quasi-Biennial variability, with amplitude peaks observed around 90 and 100 km. Comparing the DW1 amplitudes in MLT
with the zonal wind, the result reveals that the variations in DW1 amplitude correspond to the zonal wind between 20 and 30
hPa. The amplitude of DW1 is stronger during the QBO westerly wind phase than during the QBO easterly wind phase. This
result is consistent with Garcia (2023) that the wind fields of QBO at altitudes below 27 km are clearly correlated with the
DW1 amplitude. Accordingly, in this work, the zonal wind between 20 and 30 hPa is used as the criterion for defining the
QBO for DW1.
During February–May 2016 and January–May 2020, two QBO disruption events occurred (Wang et al., 2023). As shown in
Figure 1f, the phenomenon ranges from 40 to 15 hPa in 2016 and from 40 to 20 hPa in 2020, which is consistent with previous
work (Anstey et al., 2021; Newman et al., 2016). Notably, the disruption region coincides with the QBO criterion altitude for
DW1. To evaluate how the DW1 exhibits responses to the events, the corresponding time intervals are highlighted with vertical
green dashed lines. In the stratosphere (Figure 1d), within the disruption periods, amplitude enhancements are observed below
40 km compared to other QBO easterly phases. Similarly, in the MLT region, the DW1 amplitudes show responses to these
events (Figure 1b). As shown in Figures 1a and 1b, DW1 amplitudes above 70 km are stronger during these disruption events
than during other QBO easterly phases, though they remain weaker than those observed during the QBO westerly phase. This
enhancement is particularly evident around 90 and 100 km.
SD-WACCM-X simulations reproduce the SABER observations of DW1 remarkably well in response to QBO disruptions. In
Figures 1a, 1b, 1f, and 1g, both datasets show enhanced amplitudes during the February–May 2016 and January–May 2020
events. The difference arises in vertical structure and magnitude. Above 70 km, SABER exhibits three distinct DW1 peaks
near 80, 90, and 100 km, whereas SD-WACCM-X shows a single peak at approximately 108 km. In the stratosphere above
40 km, both model and observations peak at similar altitudes, but the simulated amplitudes remain weaker than SABER result.
Below 40 km, the model captures the QBO-modulated DW1 seen in Figures 1c, 1d, 1i, and 1j. These discrepancies likely stem
from the MERRA-2 nudging applied up to ~50 km in SD-WACCM-X. In this nudged region, DW1 comprises both propagating
and non-propagating components (Garcia, 2023; Chapman & Lindzen, 1970). Sakazaki et al. (2018) showed that MERRA-2
may underestimate the contribution of the non-propagation mode of DW1 (Figure 4 in that work). This feature may explain
why the amplitude of DW1 is lower than that in SABER and the complex variation of SABER above 40 km.
To assess the DW1 response to QBO disruption events over a broad latitude range, the differences between QBO disruption
and regular QBO easterly and westerly phases are calculated. The DW1 amplitudes used are the result after 13 months low-
pass filtering. Since the DW1 amplitudes typically peak between February and April each year (e.g., Xu et al., 2009; Mukhtarov
et al., 2009; Garcia, 2023), only the amplitudes during these three months are considered. The classification method for
different QBO phases is as follows. Regular QBO phases were classified using the following method. QBO westerly phase
(QBOW): February–April zonal wind at 20 hPa is continuously westerly, or zonal wind at 30 hPa is westerly while 20 hPa
undergoes an easterly-to-westerly transition. Easterly phase (QBOE): any remaining cases. The selection of regular QBO
phases is limited to data from 2002 to 2014, as QBO disruption events occurred after 2015. Additionally, since observations
in 2002 are mainly available from March to April, data from this year are excluded. The years 2004, 2006, 2008, 2011, 2013,
and 2014 are classified as QBOW; 2003, 2005, 2007, 2009, 2010, and 2012 as QBOE. For each phase, all filtered amplitudes
across the selected months are averaged, while processing data for 2016 and 2020 separately. This approach enables a direct
comparison of DW1 amplitude anomalies in both latitude and altitude between disruption and regular QBO conditions.

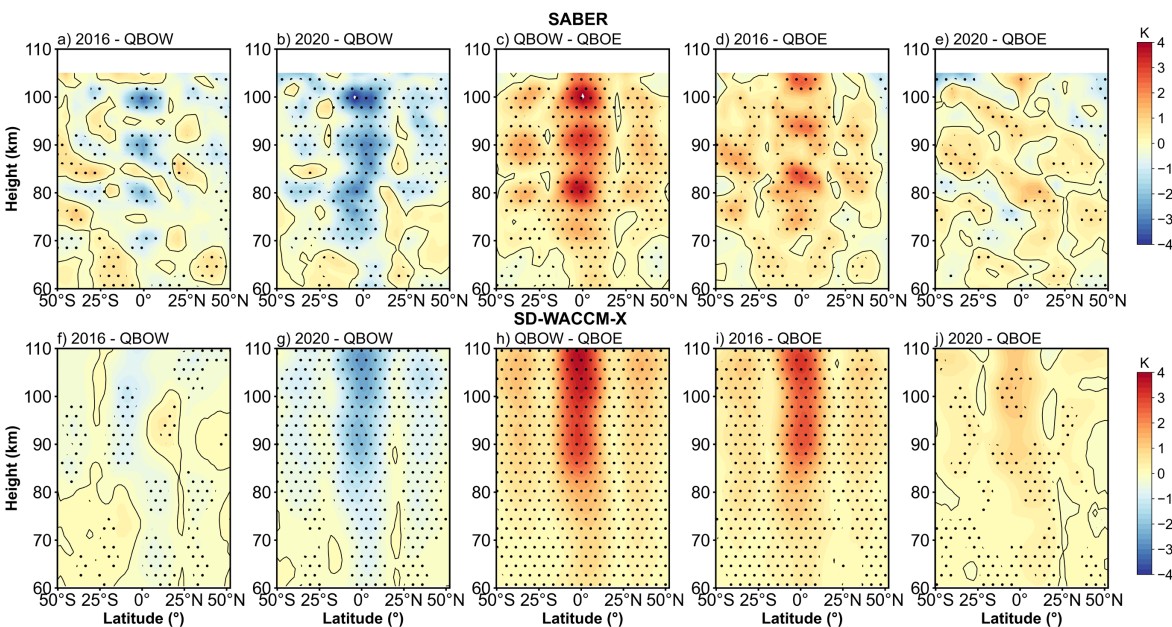


**Figure 2. Amplitude differences of the DW1 after low-pass filtering between different QBO phases in the mesosphere and lower**
**thermosphere (MLT) as a function of latitude and altitude. The difference is based on the average from February to April. (a-e) are**
**corresponding to the difference from the 2016 disruption event minus QBO westerly phases (2016-QBOW), 2020 disruption event**
**minus QBO westerly (2020-QBOW), QBO westerly minus QBO easterly (QBOW-QBOE), 2016 disruption event minus QBO**
**easterly (2016-QBOE) and 2020 disruption event minus QBO easterly (2020-QBOE). (f-j) is similar to (a-e) but for SD-WACCM-X**
**simulation result. The black lines indicate the zero lines. The dotted areas indicate the differences that are significant at the 95%**
**confidence level.**

Figure 2 gives the difference in DW1 amplitudes during various QBO phases in the MLT region. The significance of the
differences was assessed using Welch's t-test, and values exceeding the 95 % confidence threshold are highlighted in dotted.
The five columns correspond to the 2016 disruption event minus QBO westerly (2016-QBOW), 2020 disruption event minus
QBO westerly (2020-QBOW), QBO westerly minus QBO easterly (QBOW-QBOE), 2016 disruption event minus QBO
easterly (2016-QBOE) and 2020 disruption event minus QBO easterly (2020-QBOE), respectively. The relative change
between different QBO phases is also calculated (e.g., $\frac{QBOW-QBOE}{QBOE}$, and so on). The comparison between QBOW and QBOE
(Figure 2c) reveals that DW1 amplitudes are significantly larger during QBOW, particularly at the equator and around 30°N/S
above ~75 km. The enhancements reach ~2.79 K (~34.5 %) at the equator and ~0.79 K (~20.6 %) at 30°N/S, with peak values
as high as ~3.30 K (~38.5 %) and ~1.19 K (~31.7 %) at respective latitudes. During the 2016 disruption (Figures 2a, 2d), DW1
amplitudes lie between QBOE and QBOW values. The clear enhancement could be found from 75 km to 105 km. The pattern
in 2016–QBOE closely resembles that of QBOW–QBOE, although the equatorial peaks appear at slightly higher altitudes.
The enhancements reach ~1.56 K (~20.5 %) at the equator and ~0.54 K (~14.4 %) at 30°N/S. The peak enhancements relative
to QBOE reach ~2.40 K (~26.5 %) at the equator and ~0.87 K (~29.5 %) at 30°N/S. Compared to QBOW, however, the
difference drops to –2.28 K (–18.8 %) at equator and ~0.12 K (4.6 %) at 30°N/S. In contrast, the 2020 disruption event shows
weaker amplitude increases relative to QBOE (Figures 2b, 2e). The clear enhancement occurs from 75 km to 90 km. The
increment reaches ~0.50 K (~6.0 %) at the equator and ~0.26 K (~7.7 %) at 30°N/S, with a peak enhancement of only ~0.91
K (~11.6 %) at the equator and ~0.31 K (~14.2 %) at 30°N/S. Compared to QBOW, the difference is -2.3 K (~-21.1%) at the
equator and -0.57 K (~-12.0 %) at 30°N/S. These values are considerably lower than those observed during the 2016 event or
the typical QBOW enhancement. The SD-WACCM-X model reproduces the general features described above (Figures 2f–2j),
though the vertical structure of the simulated amplitudes differs slightly from observations.

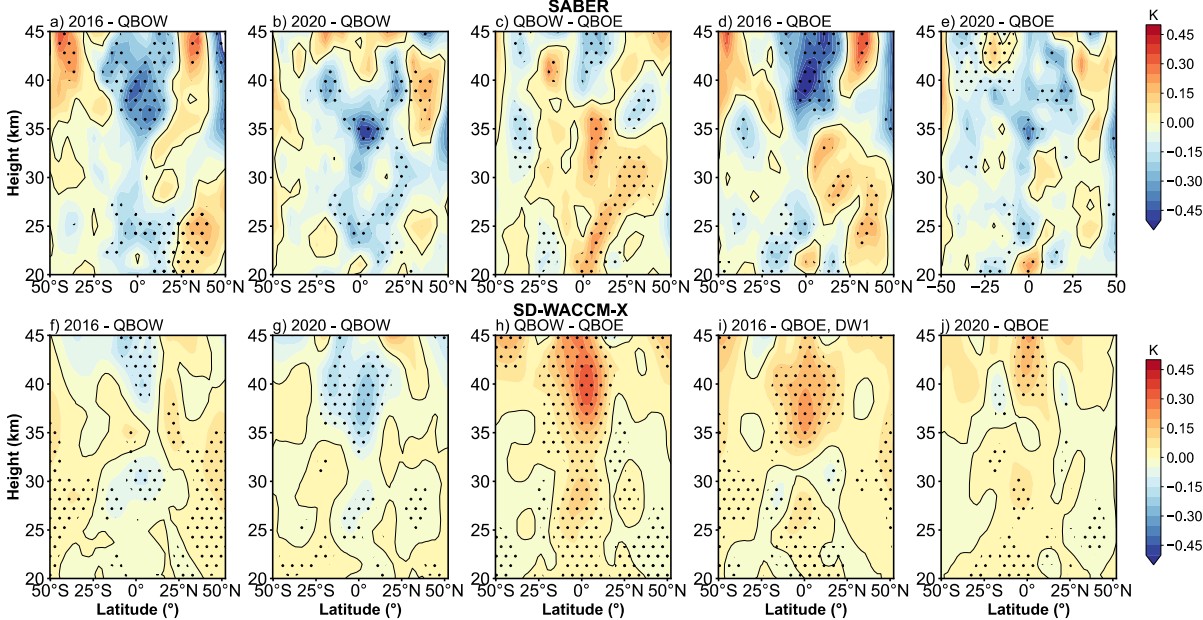


**Figure 3. Similar to figure 2 but in stratosphere. (a-e) give the difference result derived from SABER. (f-g) give the difference result**
**derived from SD-WACCM-X.**

Figure 3 compares the stratospheric DW1 amplitude differences derived from the SABER dataset and SD-WACCM-X
simulations. The enhancement pattern resembles that seen in the MLT region but is confined to tropical latitudes. Because
SABER exhibits complex variability above 40 km, the analysis is restricted to altitudes below that level. As shown in Figure
3c, the DW1 amplitudes during QBOW exceed those during the QBOE by ~0.21 K (~37.9 %) at around 20-25 and 30-35 km.
In SD-WACCM-X result (Figure 3h), the positive peaks are found at 25-30 km and 35-40 km, which is ~0.21 K (~27.4 %).
The amplitudes during the disruption events are much weaker relative to those during QBOW phases shown in both datasets
(Figure 3a, 3b, 3f and 3g). Compared to QBOE, the strengthening during the 2016 QBO disruption event occurs at
approximately 30–35 km in SABER (Figure 3d) and 35-45 km in SD-WACCM-X (Figure 3i), which is ~0.15 K (~21.8 %)
and 0.20 K (~23.9 %), respectively. During the 2020 event, the amplitudes are comparable to those during QBOE (Figure 3e
and 3j).

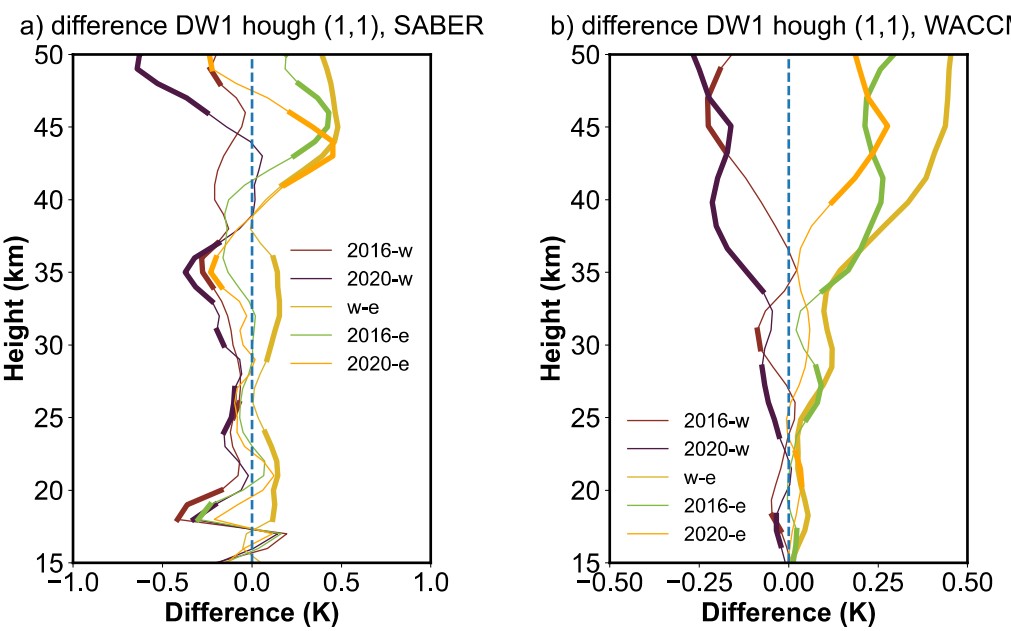


**Figure 4. Amplitude differences profiles of the DW1 (1, 1) mode after low-pass filtering between different QBO phases in the stratosphere like Figure 2. (a) gives the difference result derived from SABER. (b) give the difference result derived from SD-WACCM-X. The bold lines indicate the differences that are significant at the 95% confidence level.**


Figure C2 presents the low-pass time series of the equatorial DW1 amplitude and the (1,1) mode amplitude at 95 km, showing
that the (1,1) mode closely follows the equatorial DW1 amplitude. In the stratosphere, however, the superposition of
propagating tides and trapped modes complicates the interpretation. To separate these contributions, Figure C1 compares the
amplitudes of the (1,1) and (1, –2) modes under different QBO phases. The trapped mode is dominant below 60 km, while the
(1,1) mode is relatively weaker. A clear distinction between QBOW and QBOE is evident in the (1,1) mode (Fig. C1a), whereas
the (1, –2) mode shows little difference between the phases. Together, Figures C1 and C2 indicate that the (1,1) mode captures
nearly all of the QBO-related variability in the MLT region, motivating a closer examination of the (1,1) mode in the
stratosphere.
Figure 4 shows the vertical profiles of amplitude differences in the DW1 (1,1) mode between QBO phases after low-pass
filtering. The bold lines denote differences significant at the 95% confidence level. In SABER observations (Fig. 4a),
amplitudes during QBOW exceed those in QBOE throughout 20–45 km. During the 2016 and 2020 events, amplitudes remain
close to those in QBOE between 20–40 km but become stronger above 40 km, with maximum differences of ~-0.15 (~-11 %),
-0.18 (~-12 %), ~0.36 K (~36 %), ~0.21 K (~21 %), and ~0.18 K (~17 %) for 2016-QBOW, 2020-QBOW, QBOW–QBOE,
2016－QBOE, and 2020–QBOE, respectively. WACCM-X simulations (Fig. 4b) reproduce a similar vertical pattern: during
the disruption events, amplitudes lie between QBOE and QBOW values in the 20–50 km region.
**3.2    DW1 phases response to QBO disruption events**
In this section, whether the DW1 phases and wavelengths respond to QBO disruptions will be analysed. As discussed above,
the DW1 QBO variability is mainly in (1, 1) mode. Hence, we focus on the phase of (1, 1) mode. As noted previously, the
pronounced DW1 amplitude observed from February to April renders the phase during this period an important variable. Hence,
the statistics are based on these periods. Since the phase values change cyclically (e.g., it jumps from pi to -pi), causing the
overestimation of the standard deviation. We apply the following method. We first calculate averages and standard deviation
(or error) of sine and cosine Fourier components, and then calculate the average phase and its confidence interval using error
propagation. The mean value and its 95% confidence interval in different QBO phases (listed in section 3.1) are calculated.
The statistical results for the phases in 2016 and 2020 are calculated separately.

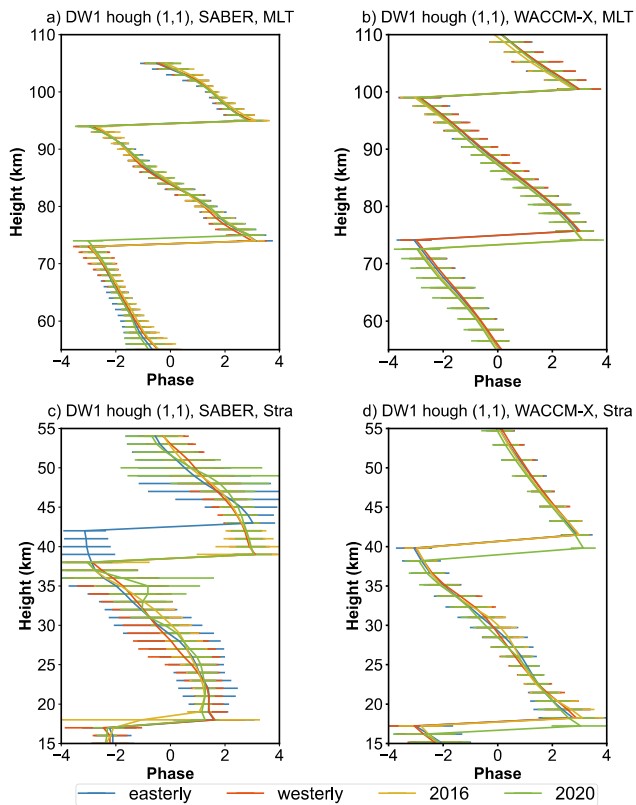


**Figure 5. The DW1 (1, 1) mode vertical phase structure in mesosphere and lower thermosphere (MLT) and stratosphere averaged**
**from February to April during QBO westerly phase (orange), QBO easterly phase (blue), 2016 QBO disruption event (yellow) and**
**2020 QBO disruption event (green). (a, c) give the SABER observation result. (b, d) give the WACCM-X simulations. The error bar**
**denotes the 95% confidence interval of the phases for each height.**

Figure 5 illustrates the vertical phase structure of DW1 (1, 1) mode in the mesosphere and lower thermosphere (MLT) and
stratospheric regions, respectively, averaged over the February–April period. The results are presented for various QBO phases
at different latitudes, based on data from (a, c) SABER and (b, d) SD-WACCM-X. Error bars indicate the 95% confidence
interval of the phase average. The lines represent different QBO phases and events: QBO westerly phase (orange), QBO
easterly phase (blue), the 2016 QBO disruption event (yellow), and the 2020 QBO disruption event (green).
In the MLT region (Figure 5a and 5b), the vertical phase profiles exhibit minimal differences across the QBO westerly, easterly
and 2016 phases. The structures are nearly identical in both the simulations and observations, with two phase peaks
(approximately $\pi$ rad) consistently present. The peak altitudes remain almost unchanged among the different QBO phases,
suggesting a limited phase response to QBO disruption events in the MLT region. During the 2020 event, the phase peaks at
around 75 km is higher than that during other QBO phases in SABER and lower than that in WACCM-X.

The DW1 vertical phase structures in the stratosphere region are given in Figure 5c and 5d. In SABER observations, there are clear differences between QBOW and QBOE. The phase peaks (at around 40 km) during the QBOW occur about 3 km lower than those during QBOE. During the QBO disruption events, the phase structure is similar to that during the QBOW. From WACCM-X simulations, the feature is similar to the pattern in MLT (Figure 5b). During the 2020 disruption event, the phase peaks are lower than other QBO phases by about 1 km.

The phase peaks described above (~π rad) are used to calculate the DW1 wavelengths in both the stratosphere and MLT regions. The altitude difference between the two peaks is taken as the wavelength, following the method of Liu et al. (2021). The statistical results of DW1 (1, 1) mode wavelengths under different QBO phases are summarized in Table 1, which lists the mean values and standard deviations at various altitudes. In the MLT region, the mean wavelengths are ~21 km in the SABER dataset and ~25 km in the SD-WACCM-X dataset. The wavelengths during QBO disruption events are comparable to those during the QBO westerly and easterly phases, a feature also captured in the SD-WACCM-X simulations. In the mesosphere, the mean wavelengths are ~34 km in SD-WACCM-X and ~33 km in SABER. In this region, there are clear differences between QBOW and QBOE. The QBOE wavelength is about 2 km shorter than that during QBOW. In the stratosphere, the QBOE wavelength is about 2 km longer than that during QBOW. The wavelengths during the QBO disruptions are close to those during QBOW.

According to the theoretical framework proposed by Forbes and Vincent (1989) and Kogure and Liu (2021), zonal winds modify the intrinsic frequency of tides through Doppler shifting, thereby altering their vertical wavelengths. Specifically, westerly winds lead to longer DW1 vertical wavelengths, whereas easterly winds result in shorter ones. However, the dependence shown in Table 1 differs from that reported in previous studies. This discrepancy can be attributed to differences in methodology. In this study, the vertical wavelengths are determined from the phase difference between adjacent peaks (+π). In the stratosphere, one of these peaks typically occurs in the lower stratosphere (~18 km) and the other in the upper stratosphere (~40 km). Consequently, the estimated wavelengths encompass the combined influences of both the QBO and SAO, producing a mixed result that deviates from earlier findings.

**Table 1. The comparison of mean (left of the slash) and standard deviations (right of the slash) of DW1 (1, 1) mode wavelengths (in km) revealed by SD-WACCM-X and SABER from 15 km to 105 km between QBO westerly phase, easterly phase, 2016 disruption event and 2020 disruption event calculated from February to April.**

| Data | SD-WACCM-X | | | SABER | | |
|---|---|---|---|---|---|---|
| altitude | ~15 km – ~ 40 km | ~40 km – ~ 75 km | ~75 km – ~105 km | ~15 km – ~ 40 km | ~40 km – ~ 75 km | ~75 km – ~105 km |
| Westerly | 22.97/1.49 | 34.47/1.79 | 25.10/1.84 | 21.81/1.44 | 33.12/1.78 | 21.29/1.04 |
| Easterly | 22.51/1.73 | 34.42/2.15 | 25.60/2.20 | 24.46/1.99 | 30.84/2.35 | 20.56/1.30 |

| | | | | | | |
|---|---|---|---|---|---|---|
| 2016 | 22.56/1/33 | 33.26/1.58 | 25.58/2.03 | 21.48/2.31 | 33.32/2.10 | 21.28/0.85 |
| 2020 | 22.71/1.87 | 33.80/2.68 | 26.27/2.41 | 21.08/1.77 | 34.24/1.46 | 20.39/1.35 |


## 4 Discussion

In this section, we discuss how the QBO disruptions modulate the DW1 by several mechanisms from the lower atmosphere to
the upper atmosphere. As in Introduction, three primary mechanisms are considered: background zonal wind and its latitudinal
shear (e.g., Forbes and Vincent, 1989; Hagan et al., 1999; McLandress, 2002b); diurnal heating (McLandress, 2002b; Riggin
and Lieberman, 2013; Ortland, 2017); tide–gravity wave (GW) interactions (Mayr et al., 1998; McLandress, 2002a; Lu et al.,
2012; Wang et al., 2024). The mechanism analysis will be organized by tidal heating (troposphere and stratosphere), the
dissipation and tidal propagation (from stratosphere to mesosphere) and tide-gravity wave interaction (mesosphere and lower
thermosphere).

### 4.1 Tidal heating variation during the QBO disruption events

The excitation sources of DW1 can be broadly classified into three categories: (1) solar radiation in the near-infrared (IR)
absorbed by tropospheric $H_2O$, (2) solar radiation in the ultraviolet (UV) absorbed by stratospheric and lower mesospheric $O_3$,
and (3) solar radiation absorbed by $O_2$ in the Schumann–Runge bands and continuum (Hagan, 1996). Additionally, Kogure
and Liu (2021) highlighted the role of latent heating in modulating DW1. It is worth noting that the timing of the 2016 QBO
disruption event coincides with the phase of the extreme El Niño (e.g., Santoso et al., 2017; Hu and Fedorov, 2017). El Niño
itself could modulate the DW1 (Kogure and Liu, 2021). Attention should be paid to the contribution of water vapor and latent
heating jointly influenced by QBO disruption and 2015/16 extreme El Niño. Overall, this section focuses on examining the
effects of water vapor radiative heating, ozone radiative heating, and latent heating on DW1.

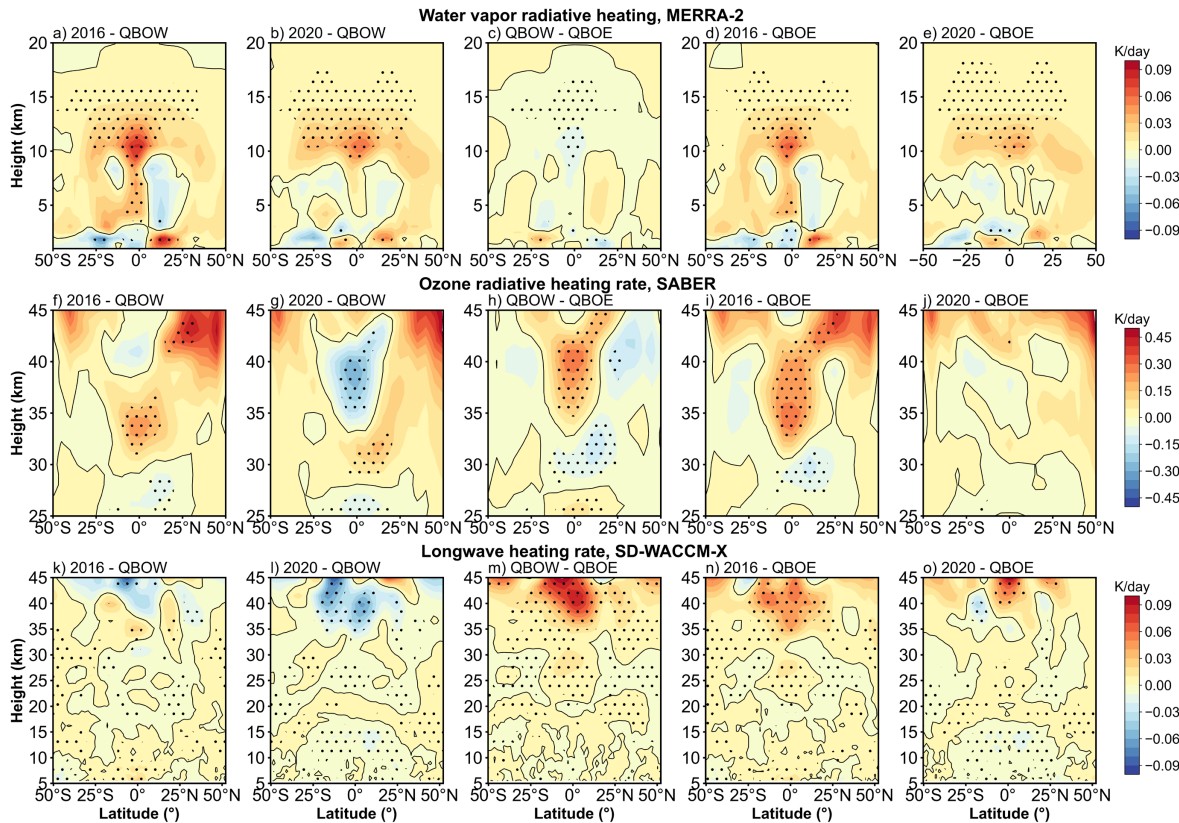

**Figure 6. As in Figure 2 and Figure 3, but the difference of amplitude in DW1 component (after low-filtering) of (a-e) water vapor heating rate DW1 component from MERRA-2, (f-j) ozone heating rate DW1 component from SABER and (k-o) longwave heating rate from SD-WACCM-X. The dotted areas indicate the differences that are significant at the 95% confidence level.**

Figure 6 presents the difference of amplitude in the DW1 component of water vapor radiative heating rate, ozone radiative heating rate, and longwave heating rate. The calculation method is consistent with the method described in Section 3.1. During QBOE and QBOW, the DW1 component of water vapor heating remains nearly unchanged (Figure 6c). However, during the 2016 QBO disruption (Figures 6a, 6d), a notable enhancement in water vapor heating appears between 10–13 km altitude around the equator. The difference between 2016 and QBOE is ~0.02 K day$^{-1}$ with increases of ~2.5% relative to QBOW. The difference between 2016 and QBOW is 0.03 K day$^{-1}$ with increases of ~3.7% relative to QBOW. A similar pattern is seen during the 2020 QBO disruption event (Figures 6b and 6e). The relative changes of the regional average increase by ~1.2 % compared to QBOW and ~2.3 % compared to QBOE.

Figures 6f–6j reveal that the largest QBO-related differences in the DW1 component of ozone heating occur near the equator between 30 and 45 km. In QBOW, ozone heating rates between 35 and 45 km exceed those in QBOE by ~2.1 % (Figure 6h). During the 2016 QBO disruption event (Figures 6f and 6i), ozone radiative heating rates are ~3.6 % larger than those in the

QBOW between 30 and 35 km and ~2.9 % larger than those in the QBOE within the 30–40 km range. In contrast, during the
2020 disruption event (Figures 6g and 6j), the ozone heating rate is comparable to that of the easterly phase and lower than
that of the westerly phase in the 35–45 km altitude range.
In the SD-WACCM-X simulation, the longwave heating rate accounts for the effects of three major absorbers: $H_2O$, $CO_2$, and
$O_3$ (Neale et al., 2010). This parameter could be used to verify the effect of the water vapor and ozone radiative heating. The
DW1 component of the longwave heating rate from SD-WACCM-X is shown in Figures 6k–6o. The heating rate difference
between the QBOW and QBOE reveals a positive peak at 40 km near the equator, with no significant difference at the
equatorial tropopause (Figure 6m). The feature corresponds to the observed pattern (Figures 6c and 6h). In the 2016 disruption
case, the simulated equatorial heating rate exhibits positive peaks around 35 km and 15 km (not significant), aligning well with
observations in terms of altitude (Figure 6k and 6n). In the 2020 disruption case, the simulation (Figure 6l and 6o) agrees with
the observed stratospheric heating features (Figures 6g and 6j). However, at around 15 km, the simulation shows negative
peaks near the tropopause, whereas the observations indicate positive peaks (Figures 6b and 6e). As longwave heating
incorporates contributions from multiple absorbers, the discrepancies may be attributed to the influence of other constituents.
As discussed above, the (1,1) Hough mode captures nearly all QBO-related variability in the MLT. Accordingly, the (1,1)
component of the ozone heating rate is extracted for diagnosis. Numerous studies have noted that the vertical thickness of
ozone heating (~40 km) is large compared with the relatively short vertical wavelength of the DW1, implying weak projection
onto the (1,1) and thus limited excitation efficiency (e.g., Chapman and Lindzen, 1970; Hagan, 1999; Garcia, 2023). Studies
with GSWM and the Tide Mean Assimilation Technique (TAMT) further indicate that DW1 forced by ozone heating tends to
be out of phase with DW1 forced by water-vapor heating, reducing the amplitudes (Hagan, 1996; Ortland et al., 2017).
Consistent with this mechanism, MLS observations show a pronounced depression of the tropical diurnal tide near 1.0 hPa
(~49.5 km; Wu et al., 1998), which may be attributed to interference between the upward-propagating (1,1) tide and a locally
forced component from ozone heating. Figure 7 compares DW1 (1,1) temperature and ozone heating rate between different
QBO phases and shows a suppressed (1,1) amplitudes feature around ~50 km, while ozone heating peaks slightly below this
level. This feature aligns with the MLS evidence. Therefore, the ozone may not play a positive role for the DW1 (1, 1) mode.
Whether ozone heating modulated DW1 (1, 1) mode requires more detailed investigation like model simulation from Kogure
et al. (2023).

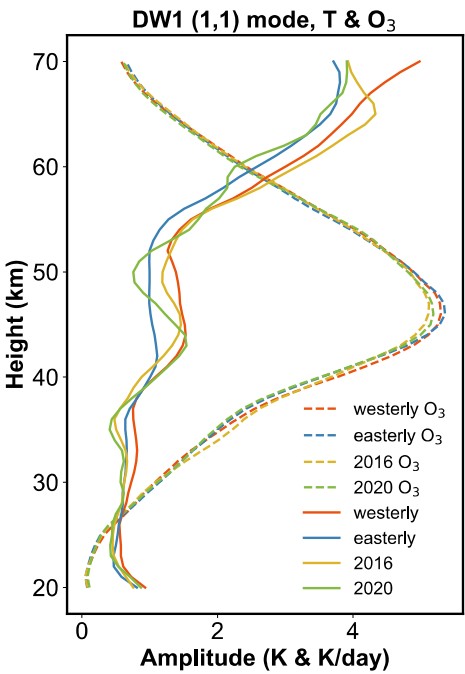

**DW1 (1,1) mode, T & O₃**


**Figure 7. The comparison between the temperature and heating rate of the DW1 (1, 1) mode between different QBO phases.**

The DW1 (1,1) mode is primarily excited by water-vapor heating (Forbes and Garrett, 1978). As shown in equation A1 and A10 in Appendix A, the concentration of water vapor is one of the key factors controlling water vapor radiation. During the 2016 QBO disruption, which coincided with the strong 2015/2016 El Niño, the two phenomena jointly modulated the DW1 water vapor heating sources. El Niño enhances moisture anomalies that increase with altitude, culminating in pronounced positive signals in the upper troposphere and lower stratosphere (UTLS) (Johnston et al., 2022). In contrast, the occurrence of the 2016 QBO disruption introduces a shear transition from westerly to easterly near 40 hPa, which strengthens tropical upwelling and lowers cold-point temperatures. This dynamical response injects $H_2O$-poor air into the lower stratosphere, partially offsetting the El Niño–driven moistening. The water vapor concentrations remain above the climatological seasonal cycle under the modulation of these two phenomena (Diallo et al., 2018). Unlike 2016, the 2020 disruption produces only weak lower-stratospheric dehydration (~2–3 %) because enhanced upwelling and cold-point cooling are suppressed. Instead, anomalously warm tropopause temperatures associated with Australian wildfire smoke facilitate significant moistening of the lower stratosphere (Diallo et al., 2022). It is foreseeable that the increase in water vapor concentration modulated by QBO disruptions and 2015/16 El Niño event will lead to an increase in the radiative heating rate of water vapor.

Figure 8 presents the water vapor radiative heating rate profiles of the DW1 (1,1) mode for different QBO phases and their differences. The heating rate exhibits large values in the troposphere, extending up to ~10 km. The average magnitude could reach ~0.62 K day⁻¹. During the 2016 QBO disruption event (Fig. 8b), the maximum difference occurs at 10.5 km, reaching 0.043 K day⁻¹, which represents an ~8 % increase relative to QBOE. However, the DW1 amplitude varies by ~20.5 % compared

to QBOE, indicating that water-vapor heating accounts for only ~39 % of the observed amplitude difference. This feature
suggests that additional mechanisms must be involved. A similar enhancement of water-vapor heating is observed during the
2020 event, with the largest difference again at 10.5 km (~0.026 K day⁻¹), corresponding to an ~5 % increase relative to QBOE.

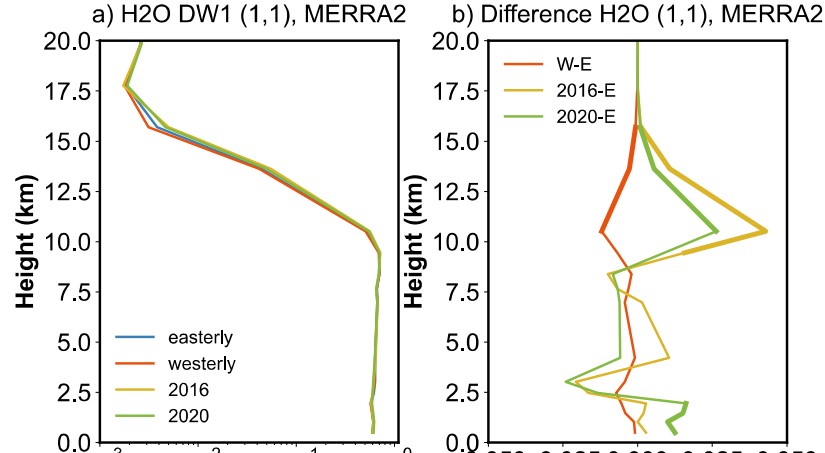


**Figure 8. Heating rate profiles of the DW1 (1, 1) mode between different QBO phases and their differences. (a, b) give the water**

**vapor heating profile and its difference derived from MERRA2. The bold lines indicate the differences that are significant at the**
**95% confidence level.**

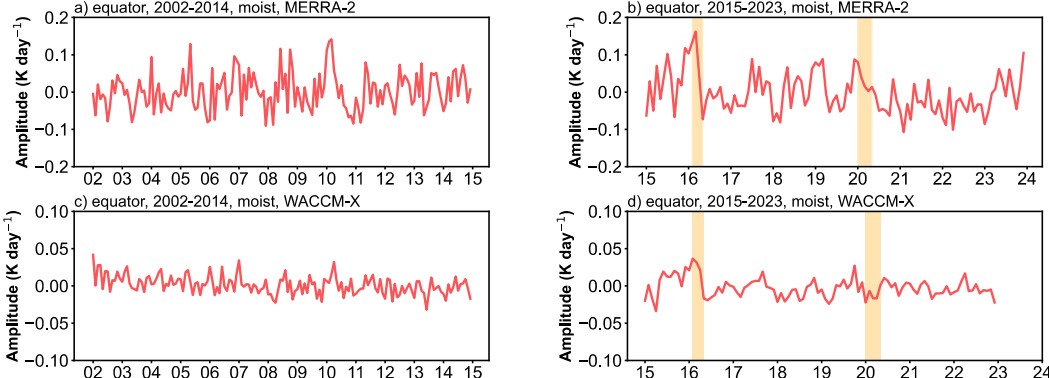


**Figure 9. (a, b) The deseasonalized time series of DW1 amplitudes of latent heating rate (K day⁻¹) at equator averaged from 800 hPa**
**to 200 hPa derived from MERRA-2. (c-d) is as in (a-b) but from SD-WACCM-X. The orange-filled areas represent two QBO**
**disruption events.**
Figure 9 shows the deseasonalized time series of the DW1 component of latent heating rate (K day⁻¹) at the equator, averaged
from 800 hPa to 200 hPa. In this tropospheric layer, the latent-heating signal shows less differences between QBOW and
QBOE phases. Therefore, deseasonalization is directly applied to the full time series without separating the two QBO states.

In MERRA-2 and SD-WACCM-X, the anomaly peaks reach 0.162 K day$^{-1}$ and 0.037 K day$^{-1}$, respectively, which correspond to increases of about ~32 % and ~25 % above their climatological means (0.50 K day$^{-1}$ and 0.15 K day$^{-1}$). When averaged over February-April in 2016, the anomalies remain elevated at 0.11 K day$^{-1}$ (~22 %) in MERRA-2 and 0.03 K day$^{-1}$ (~19.2 %) in SD-WACCM-X. In contrast, during the 2020 QBO disruption event, the amplitudes in both MERRA-2 and SD-WACCM-X remain closer to the climatological means, with deviations of 0.018 K day$^{-1}$ and -0.013 K day$^{-1}$, respectively. These results suggest that latent heating may contribute to the amplification of DW1 amplitudes during the 2016 QBO disruption event but show little effect during the 2020 event.

**4.2 Effect of zonal wind and its latitude shear during the events**

In this section, we focus on the effects of the background wind on tides during their upward propagation. As discussed in Forbes and Vincent (1989), zonal winds distort the tidal expansion functions such that they are amplified and broadened in the winter hemisphere (U > 0) but are considerably diminished under summer conditions. As shown in Figure 1f, during the 2016 QBO disruption, the westerly wind layer is unusually thick in the stratosphere, though still weaker than in the normal QBOW. In contrast, during the 2020 event, the westerly wind layer is extremely shallow, essentially indistinguishable from the easterly phase. Thus, the zonal wind in the stratosphere during the 2016 QBO disruption is conducive to the growth of tidal amplitude. During the 2020 QBO disruption event, the influence of zonal winds in the stratosphere on tides is essentially consistent with that observed during the QBOE period.

Additionally, the zonal wind influences the intrinsic frequency through Doppler shifting and therefore modifies the variation of the vertical wavenumber. The increase or decrease of the vertical wavenumber depends on the direction of zonal wind. Under the usual Newton-cooling/Rayleigh-friction parameterizations, the effective dissipation is approximately proportional to the squared local vertical wavenumber (Forbes and Vincent, 1989; Kogure and Liu, 2021). Consequently, zonal wind could also influence the dissipation process. To examine the dissipation during the event, we apply the amplitude ratio method.

As in Forbes and Vincent (1989), the amplitude growth equation is:

$$\frac{A(z)}{A(70)} = \exp\left\{\int_{70}^{z}\left[-k_i + \frac{1}{2H}\right]dz\right\} \quad (3)$$

where A is the amplitude, z is the altitude (in km), $H$ is the local scale height and $k_i$ is the imaginary part of the complex vertical wavenumber that governs the damping of the amplitude profile. Calculating the ratio of amplitude using Equation (3) during two different QBO phases (e.g., 2016/QBOE) yields:

$$\frac{A_{2016}(z)}{A_{QBOE}(z)} = \exp\left\{\int_{70}^{z} -\left(k_{i,2016}(z) - k_{i,QBOE}(z)\right)dz\right\} \quad (4)$$

The scale height term is removed, leaving the dissipation term. Thus, the changes in amplitude ratio may reflect tidal dissipation variations at the altitude.

Figure C3 presents the ratio results derived from SABER observations and SD-WACCM-X simulations. In the SABER data (Figure C3a), during the 2016 event, two distinct peaks appear in the lower stratosphere near 22 km and 30 km when comparing the disruption events with the QBOE phase (green lines), possibly indicating relatively less dissipation during the 2016 QBO

disruptions. During the 2020 event, the lower peak (~22 km) is close to that during 2016 event, while the upper peak (~30 km)
is relative weak to the 2016 event. This may suggest a relatively large dissipation at those heights. The SD-WACCM-X
simulations reproduce a similar pattern, although the peak altitudes differ slightly. All simulated ratios remain above 1, which
may indicate stronger tidal source activity in the SD-WACCM-X simulations. Overall, these results suggest that during QBO
disruptions, zonal wind may lead to relatively less dissipation processes, thereby affecting DW1 amplitudes.
In addition to zonal-mean wind effects, latitudinal shear of zonal wind in the subtropical mesosphere can modulate the seasonal
variability of the (1,1) mode (McLandress, 2002b; Mayr and Mengel, 2005; Sakazaki et al., 2013; Kogure et al., 2021; Siddiqui
et al., 2022). Large values of $|\partial u/\partial y|$ at some heights are equivalent in some sense to faster rotation, which restricts the
latitudinal band or waveguide where the diurnal tide can propagate vertically, thus reducing the tidal amplitude above by
removing tidal energy at that altitude (McLandress, 2002b; Siddiqui et al., 2022). The wind shear at 18°N/S is a typical
indicator (Kogure et al., 2021).

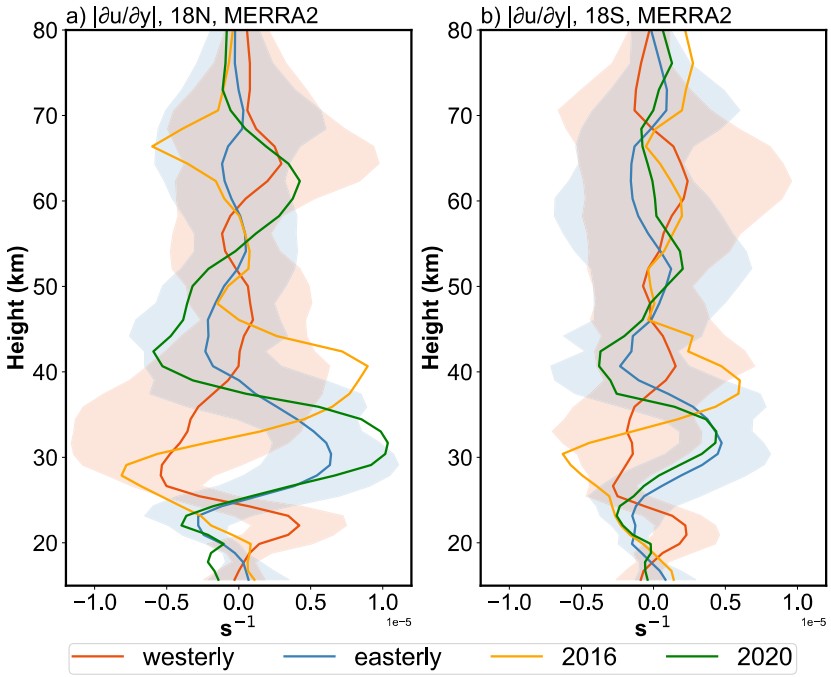

**Figure 10. The $|\partial u/\partial y|$ profiles after deseasonalized between different QBO phases at (a) 18°N and (b) 18°S. The colourful shaded**
**areas denote one standard deviation of the phases for each height.**

The monthly $|\partial u/\partial y|$ at 18°N/S is calculated, deseasonalized, and classified following the method described in Section 3.1. The
$|\partial u/\partial y|$ profiles for different QBO phases are shown in Figure 10. During QBOW, a pronounced negative anomaly appears
near 30 km, whereas during QBOE a strong positive anomaly is evident at the same altitude. During the 2016 disruption event,
the $|\partial u/\partial y|$ profile at 18°N exhibits a structure broadly similar to that of QBOW. However, from 35 to 45 km it shows strong
positive values, a feature not observed in other QBO phases. The $|\partial u/\partial y|$ profile at 18°S displays a similar vertical structure
but with smaller amplitudes. Based on this structure, the tide may be amplified near 30 km and subsequently damped near 40
km, which could partly explain why the tidal amplitudes during the 2016 disruption do not reach those observed in QBOW. In
contrast, the $|\partial u/\partial y|$ during the 2020 disruption event closely resembles the QBOE structure, suggesting that the tidal
propagation background was similar to QBOE conditions.

### 4.3 Contribution of Tide-gravity wave interaction during the events

The mesospheric diurnal tides are also affected by the interaction with GWs (Liu and Hagan, 1998; Mayr et al., 1998;
McLandress, 2002a; Li et al., 2009; Lu et al., 2012; Yang et al., 2018; Stober et al., 2021; Cen et al., 2022). These interactions
can strongly modulate tidal amplitude and phase (Liu and Hagan, 1998; Lu et al., 2009; Li et al., 2009; Wang et al., 2024). In
the mesosphere, gravity-wave drag may be linked to the QBO. As discussed in Wang et al. (2024), QBO-dependent zonal
wind shear and associated zero-wind lines filter the upward-propagating gravity waves that can reach the mesosphere, making
the gravity wave drag exhibit QBO-like features. In the tropical region of the mesosphere, due to the strong interaction between
the GWs and the semi-annual oscillation (SAO) in zonal wind, the GWs in the mesosphere exhibit a weak QBO signature.
QBO-related variations in GWs primarily exists in the mid-latitude mesosphere.
To quantify the GW forcing on the DW1, the methods of Yang et al. (2018) and Cen et al. (2022) are applied. The equation is:

$$\text{GW}_{\text{forcing}} = \text{GW}_{\text{drag}} \cdot \cos\left(\omega \cdot \left(\phi_{\text{GW}} - (\phi_T - 6)\right)\right) \tag{5}$$

Where the $\text{GW}_{\text{drag}}$ is the DW1 amplitude of GW drag, $\omega$ is the $24/2\pi$, $\phi_{\text{GW}}$ is the DW1 phase of GW drag while $\phi_T$ is DW1
amplitude of temperature.

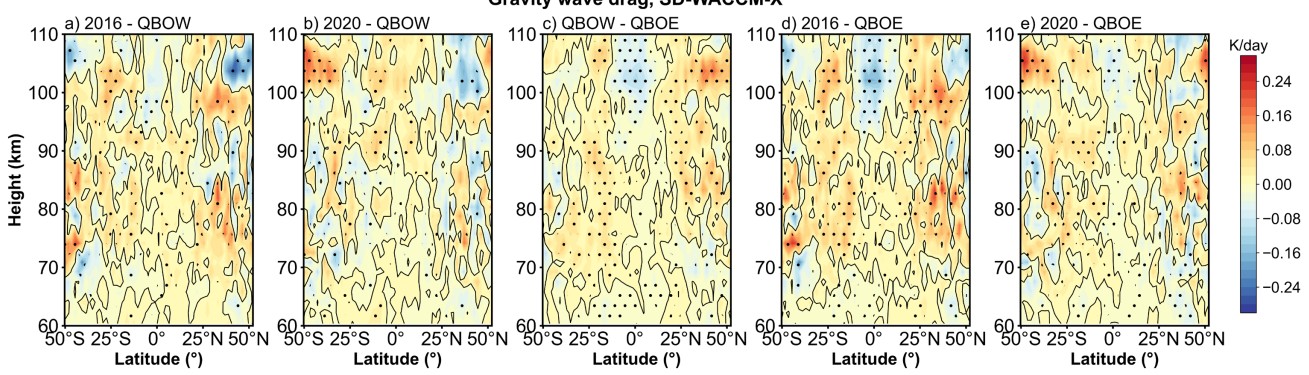

**Figure 11. Similar to figure 2 but the difference of gravity wave forcing. (a-e) give the difference result derived from SD-WACCM-X.**

After calculating the GW forcing, the classification method described in Section 3.1 is applied. As shown in Figure D1, GW
tends to damp the DW1 amplitude at nearly all latitudes above 105 km. Below ~105 km, the GWs tend to damp the DW1
amplitude at the equator while enhancing it at subtropical latitudes. Differences exist in the amplitude of gravity wave drag
between different QBO phases. Figure 11 shows the differences in GW forcing between QBO phases, with dots indicating
regions exceeding the 95% significance level. During QBOW (Fig. 11c), the equatorial damping and subtropical enhancement
are stronger than during QBOE. During the 2016 QBO disruption event, the pattern closely resembles the QBOW–QBOE
difference but exhibits a larger magnitude than QBOW (Fig. 11a). During the 2020 disruption event, the GW drag is similar
to QBOW conditions and is stronger than in QBOE. These results suggest that GW forcing exerts a significant influence on
the modulation of DW1 amplitudes across QBO phases and disruption events.
**5 Summary**
In this work, the response of global DW1 amplitudes and phases during QBO disruption events is investigated using SABER
observation, MERRA-2 dataset and SD-WACCM-X simulation results from 2002 to 2023. Additionally, the underlying
mechanisms associated with these events are explored. The findings are summarized as follows:
(1) There are clear differences in (1, 1) mode vertical phase structure and wavelengths between QBO westerly phases and
easterly phases. The DW1 (1, 1) mode vertical phase structure and wavelengths during these two QBO disruption events are
similar to those during QBO westerly phases.
(2) During the 2016 QBO disruption event, DW1 amplitudes are markedly enhanced relative to regular QBOE but remain
lower than those during QBOW. In the mesosphere and lower thermosphere (MLT), the amplitudes increase by ~1.56 K
(~20.5 %) at the equator and ~0.54 K (~14.4 %) at 30° N/S yet are smaller than those during QBOW by ~1.22 K (-10.2 %) at
the equator. A pronounced difference of the DW1 (1, 1) mode relative to QBOW and QBOE is also evident in the stratosphere,
with amplitudes ~0.21 K (~21%) higher than during QBOE and ~0.15 K (~10.9%) weaker than during QBOW.
By contrast, the 2020 disruption shows only a modest rise in DW1 amplitude relative to the regular QBOE and remain much
weaker than during QBOW. In the MLT, the amplitudes rise by ~0.50 K (~6.0 %) at the equator and ~0.26 K (~7.7 %) at
30°N/S compared to QBOE, but are smaller than those during QBOW by 2.3 K (~21.1%) and 0.57 K (~12.0%), respectively.
In the stratosphere, the amplitudes are about 0.18 K (~17.0%) larger than during QBOE but ~0.18 K (~12.5%) lower than
during QBOW.
(3) During the 2016 event, the stronger water vapour radiative heating (~8.3 % relative to QBOE and ~10.9 % relative to
QBOW) and latent heating (22% relative to both QBO phases) enhance the tides at their source region. The enhanced water
vapour radiative heating is jointly modulated by 2016 QBO disruption and 2015/16 El Niño event, whereas the enhanced latent
heating is mainly due to the 2015/16 El Niño event. Weak dissipation (zonal winds) and less tidal energy removal (zonal wind
latitudinal shear) during the tide propagate upward in the lower stratosphere do not tend to suppress DW1 amplitudes, while
gravity waves strengthen DW1 in the subtropics and damp it at the equator. Nevertheless, a stronger shear near ~40 km likely
prevents DW1 amplitudes from reaching the levels observed during normal QBO westerly phases. Overall, the joint
modulation of water-vapor radiative heating, latent heating, weak dissipation, weak energy removal and positive GW drag
contribute to a significant increase in DW1 amplitudes.
In contrast, during the 2020 event, only water vapour radiative heating exhibits a clear rise (~5 %). The dissipation and the
tidal energy removal in the stratosphere become larger, effectively suppressing DW1 enhancement. The gravity-wave effect
was weaker than in 2016 but still stronger than in QBOE. Consequently, the combined influence of water-vapor radiative
heating and GW drag contribute to a slight increase in DW1 amplitudes relative to QBOE.
This work analyzes how the DW1 varies when the highly unusual wind of QBO occurs. This phenomenon which is found in
responses at different atmospheric layers suggests an atmosphere coupling process. The observations and model simulations
give clear evidence of the connection. The possible link between the lower atmosphere trace gases variation and MLT dynamic
features is shown during these unique events. The result gives a window for exploring the mechanism of the coupling,
providing a basis for future research on the underlying mechanisms.
**Appendix A: approach for calculating the water vapor radiative heating rate**
The heating rate for water vapor mainly follows the method from Groves et al. (1982) and Lieberman et al. (2003).
As mentioned in equation 1, the heating rate could be categorized into clear sky and cloudy sky. The equation of clear
sky is given by Lacis and Hansen (1974):

$$J_{clr} = q\eta^c S_0 \cos\zeta \left[ MA(y) + \frac{5}{3} RA(y') \right] \qquad (A1)$$

with q is water vapor mixing ratio (specific humidity), $\eta$ is defined as $p/p_0$, c is defined as $0.75 - \Gamma R_M/2g$. $\Gamma$ is the
vertical lapse rate, which is 6.5K km$^{-1}$. $R_M$ is the gas constant for air. g is the acceleration of gravity. $S_0$ is the solar
constant, which is 1353 W m$^{-2}$. $\zeta$ is the solar zenith angle, the equation is:

$$\cos\zeta = \sin\theta \, \sin\delta + \cos\theta \, \cos\delta \, \cos t' \qquad (A2)$$

with $\theta$ is the latitude, $\delta$ is the solar declination. $t'$ is given by following equation:

$$t' = \lambda + \Omega t \qquad (A3)$$

with $\lambda$ is longitude in radiance, $\Omega$ is the angular frequency of Earth's rotation. $t$ is the universal time.
M is given by equation:

$$M = \frac{35}{(1224 \cos^2\zeta + 1)^{\frac{1}{2}}} \qquad (A4)$$

A(y) is given by equation:

$$A(y) = 2.9 \left[ \frac{0.635 + 0.365Y}{(Y^{0.635} + 5.925y)^2 Y^{0.365}} \right] \text{cm}^2\text{g}^{-1} \qquad (A5)$$


with:

$$Y = 1 + 141.5y \qquad (A6)$$

and

$$y = M\overline{w} \qquad (A7)$$

and

$$y' = M\bar{w}_t + \frac{5}{3}(\bar{w}_t - \bar{w}) \tag{A8}$$

The $\bar{w}$ is the effective water vapor amount, is given by equation:

$$\bar{w} = \int_z^\infty q\rho(p/p_0)^{.75}(T_0/T)^{1/2}dz \tag{A9}$$

Where $\rho$ is the air density. $\bar{w}_t$ is the total water vapor above the reflecting surface.
The cloudy sky heating rate is given by Groves (1982):

$$J_{cld} = q\eta^c S_0 \cos \zeta Z \tag{A10}$$

with Z is parameter given by:

$$Z = \sum_i \{ak'[\cosh(\xi_0 + \beta - \xi)) - \cosh(\xi_0 + \beta' - \xi)]/\sinh(\xi_0 + \beta)\}_i \tag{A11}$$

with $\xi$ is given by:

$$\xi = k'\bar{w} \tag{A12}$$

$$k' = \frac{5}{3}\alpha(\sigma + k) \tag{A13}$$

with $\alpha$, $\beta$ and $\beta'$:

$$\alpha = (1 - \omega)^{\frac{1}{2}}(1 + \omega - 2\omega f)^{\frac{1}{2}} \tag{A14}$$

$$\beta = \frac{1}{2}\ln\{[1 + \alpha - \omega f - R\omega(1 - f)] \div [1 - \alpha - \omega f - R\omega(1 - f)]\} \tag{A15}$$

$$\beta' = \beta + \frac{1}{2}\ln\left[\frac{1 - \alpha - \omega f}{1 + \alpha - \omega f}\right] \tag{A16}$$

with single scattering albedo:

$$\omega = \frac{\sigma}{\sigma + k} \tag{A17}$$

where $\sigma = 40$ cm$^{-1}$, f is 0.925, k and a are given by table 2 from Somerville et al. (1974).
**Appendix B: approach for calculating the ozone radiative heating rate**
The heating rate for ozone mainly uses the equations from Strobel/Zhu model (Strobel, 1978; Zhu, 1999) and processing
method from Xu et a. (2010). The Chappius, Hartley and Huggins bands are as follow:

$$\frac{H_{Ch}}{[O_3]} = F_c\sigma_c \exp[-\sigma_c N_3] \tag{B1}$$

$$\frac{H_{Ha}}{[O_3]} = F_{Ha}\sigma_{Ha} \exp[-\sigma_{Ha}N_3] \tag{B2}$$

$$\frac{H_{\text{Hu}}}{[O_3]} = \frac{1}{MN_3}\{I_1 + (I_2 - I_1)exp[-\sigma_{Hu}N_3 e^{-M\lambda_{long}}] - I_2 exp[-\sigma_{Hu}N_3 e^{-M\lambda_{short}}]\} \qquad (B3)$$

The $[O_3]$ is the ozone number density while the $N_3$ is the column density of $O_3$ along the solar radiation path. For
equation B1, the $F_c$ is 370 J m$^{-2}$ s$^{-1}$, the $\sigma_c$ is 2.85 $\times10^{-25}$. For equation B2, the $F_{Ha}$ is 5.13 J m$^{-2}$ s$^{-1}$, the $\sigma_{Ha}$ is 8.7 $\times10^{-}$
$^{22}$ m$^{-2}$. For equation B3, the $I_1$ is 0.07 J m$^{-2}$ s$^{-1}$ Å$^{-1}$, the $I_2$ is 0.07 J m$^{-2}$ s$^{-1}$ Å$^{-1}$, M is 0.01273 Å$^{-1}$, $\lambda_{long}$ is 2805 Å$^{-1}$, $\lambda_{short}$
is 3015 Å$^{-1}$, $\sigma_{Hu}$ is 1.15 $\times10^{-6}$ m$^{-2}$.
For the heating rate calculation, the ozone density profiles are firstly interpolated to a uniform vertical grid with 1 km spacing
from 20 km to 105 km. Then the ozone profiles are processing into zonal mean overlapping latitude bins that are 10 degrees
wide with centres offset by 5° from 50°S-50°N. The diurnal variation of the vertical profile of the ozone heating rate in each
latitude bin is calculated using the SABER ozone density and equation B1-B3, along with the diurnal variation of solar zenith
angle for the specific latitude and day of year.
**Appendix C: The feature of DW1 (1, 1) Hough mode**

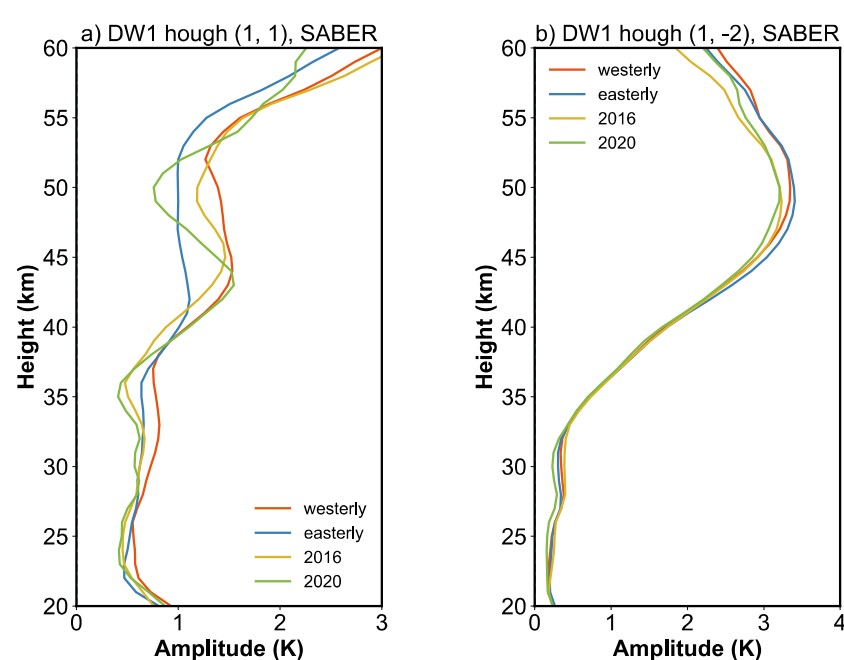

Figure C1. Amplitude profiles of DW1a) (1,1) and b) (1, -2) modes during different QBO phases

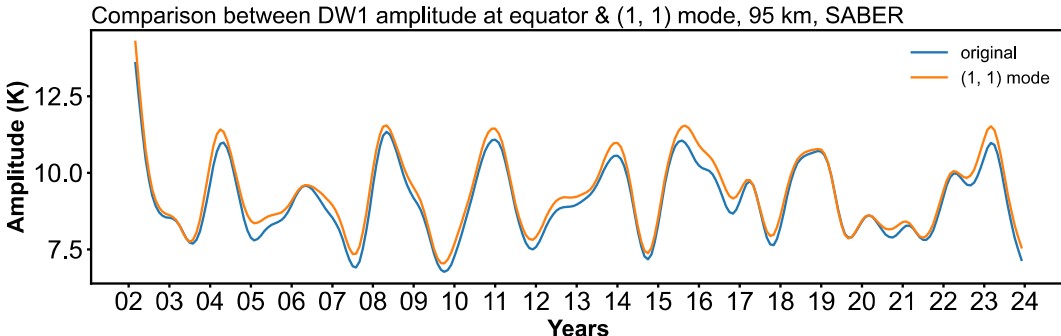

Figure C2. The amplitude time series of equatorial DW1 and (1, 1) Hough mode at 95 km.

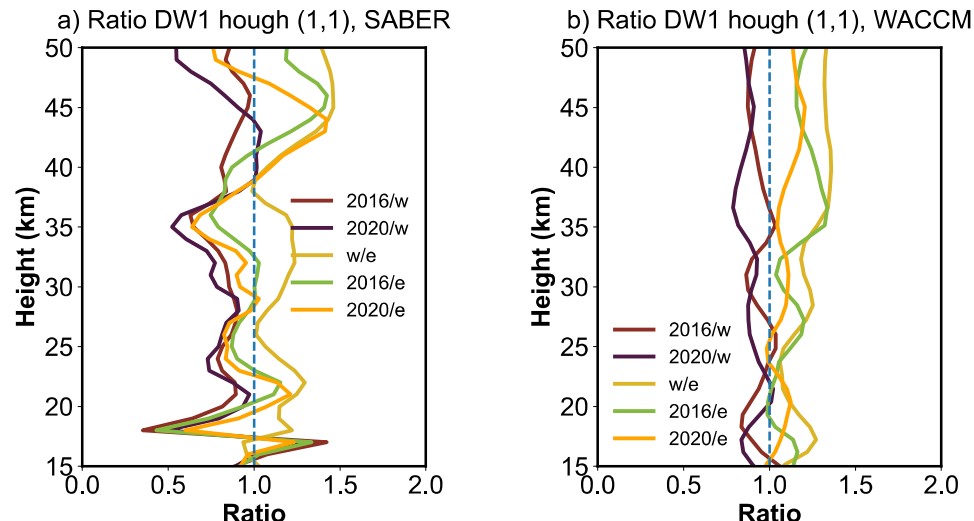

Figure C3. the vertical profile of DW1 (1, 1) mode amplitude ratio derived from 2016/westerly (dark red), 2020/westerly (dark purple), westerly/easterly (yellow), 2016/easterly (green), and 2020/easterly (orange). The dashed blue lines represent the ratio of 1.

**Appendix D: Gravity wave drag effect to DW1**

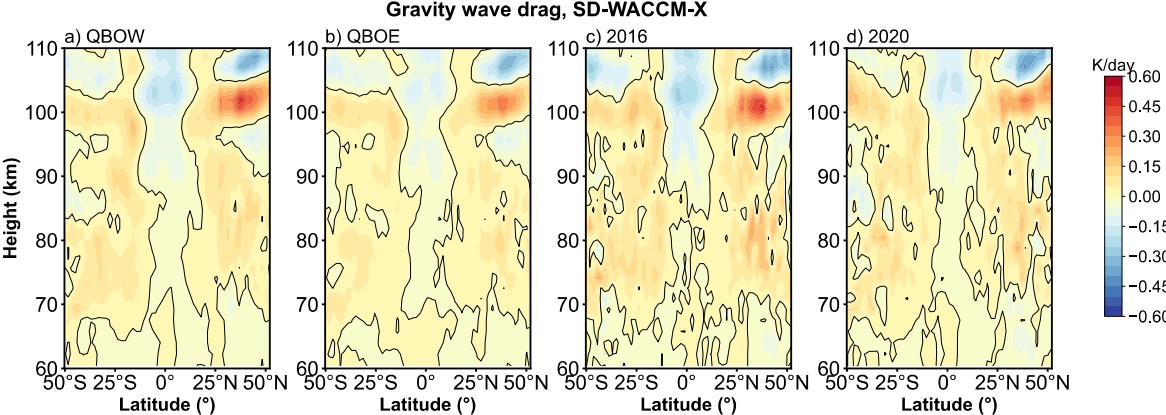

Figure D1. The gravity wave forcing on DW1 during difference QBO phases as a function of latitude and altitude

**Data availability.** SABER data is available from the SABER project data server at https://spdf.gsfc.nasa.gov/pub/data/timed/saber/. The SD-WACCM-X is retrieved from https://app.globus.org/file-manager?origin_id=d2762023-6ab4-46c9-ab12-b037cd568e42&origin_path=%2F. The QBO index is retrieved from https://acd-ext.gsfc.nasa.gov/Data_services/met/qbo/QBO_Singapore_Uvals_GSFC.txt. The Generalized Lomb-Scargle Periodogram and best-frequency fit method are provided by PyAstronomy (https://github.com/sczesla/PyAstronomy). The MERRA-2 reanalysis data can be retrieved from https://disc.gsfc.nasa.gov/datasets/M2T3NVASM_5.12.4/summary/ (zonal wind, temperature, cloud fraction, specific humidity), https://disc.gsfc.nasa.gov/datasets/M2I3NVAER_5.12.4/summary (air density), https://disc.gsfc.nasa.gov/datasets/M2T1NXRAD_5.12.4/summary (surface albedo), https://disc.gsfc.nasa.gov/datasets/M2T3NPTDT_5.12.4/summary?keywords=MERRA2%20tdt (tendency of air temperature due to moist processes).

**Author contributions.** Conceptualization: SL, GYJ; investigation: SL; methodology: SL, GYJ; project administration: BXL, GYJ and YJZ; software: SL; supervision: GYJ, BXL and YJZ; validation: BXL, GYJ and YJZ; visualization: SL; writing – original draft preparation: SL; and writing – review and editing: GYJ, BXL, XL, JYX, YJZ and WY. All authors have read and agreed to the published version of the paper.

**Competing interests.** The authors declare that they have no conflict of interest.

**Disclaimer.** Publisher's note: Copernicus Publications remains neutral with regard to jurisdictional claims made in the text, published maps, institutional affiliations, or any other geographical representation in this paper. While Copernicus Publications makes every effort to include appropriate place names, the final responsibility lies with the authors.

674

**Acknowledgements.** WACCM-X SD output data have been used in this study, and we would like to acknowledge the WACCM-X development group at NCAR/HAO for making the model output publicly available. This work was jointly supported by the Strategic Priority Research Program of the Chinese Academy of Sciences (Grant No. XDB0560000), the Pandeng Program of National Space Science Center CAS, National Key R&D program of China (2023YFB3905100), the Project of Stable Support for Youth Team in Basic Research Field, CAS (YSBR-018), the National Natural Science Foundation of China (42174212), the Chinese Meridian Project, and the Specialized Research Fund for State Key Laboratories.

682

**Financial support.** This work was jointly supported by the Strategic Priority Research Program of the Chinese Academy of Sciences (Grant No. XDB0560000), the Pandeng Program of National Space Science Center CAS, National Key R&D program of China (2023YFB3905100), the Project of Stable Support for Youth Team in Basic Research Field, CAS (YSBR-018), the National Natural Science Foundation of China (42174212), the Chinese Meridian Project, and the Specialized Research Fund for State Key Laboratories.

688

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
