# Peer review of "Migrating diurnal tide anomalies during QBO disruptions in 2016 and"

_EGUsphere, 2025_

## Author Comment (AC1)

In this revised version, three major improvements are introduced:

(1) The relationship between the 2015/16 El Niño and the 2016 QBO disruption is reviewed, with emphasis on how the El Niño event triggered the QBO disruption and how both phenomena jointly influenced the DW1 heating sources, particularly water vapor.

(2) The (1,1) mode of the DW1 is extracted and analyzed, and its amplitude and phase responses to the event are presented in Sections 3.1 and 3.2.

(3) The discussion of mechanisms is expanded. Section 4 is reorganized into three subsections addressing tidal heating, tidal propagation, and tidal–gravity wave interaction. The result of (1, 1) mode water vapor heating is given in Section 4.1. The role of ozone heating is discussed in Section 4.1. The contribution of zonal wind latitudinal shear and gravity wave drag are also discussed in 4.2 and 4.3, respectively.

**Comments 1:** I believe that the QBO disruption enhances the DW1 tide because it corresponds to the easterly phase, which is well known to amplify the tide.

Also, I wonder what differences a QBO disruption from a typical easterly QBO phase. The authors do not address this point at all. If there is no clear difference, this paper does not present a new finding, although it does confirm the QBO impact on the DW1, which is an important work. Furthermore, I question whether the enhancement of the DW1 is caused by the QBO or by El Nino.

**Response 1:** Thanks for the helpful comments. The referee gives the comment that 'I believe that the QBO disruption enhances the DW1 tide because it corresponds to the easterly phase, which is well known to amplify the tide'. The referee may misunderstand the easterly QBO phase and eastward (westerly) QBO phase. Here we give the example (Fig. 1) from Xu et al. (2009). The first panel is the amplitude of the migrating diurnal temperature tide in the MLT region at the equator and the second panel is the zonal wind in the lower stratosphere at the equator. The feature that larger-than-average diurnal tide amplitudes during the westerly phase of the QBO and smaller-than-average amplitudes during the easterly phase could be found in this figure and are supported by many works (Vincent et al., 1998; Wu et al., 2008; Xu et al., 2009; Davis et al., 2013; Araújo et al., 2017; Pramitha et al., 2021b; Garcia, 2023).

If there is no disruption event, the QBO should evolve into QBO easterly phases, which the amplitude of diurnal tides is weaker. During the period of disruption event, unique westerly wind occurs above the 40 hPa and then propagate upward nearly to 10 hPa. This QBO feature is different from the normal downward-propagating QBO westerly and easterly phases. That's why we want to investigate how the DW1 response to these unique events and reveal the possible mechanism underlying. From the figure 1 to figure 3 in manuscript, we could see the amplitude of DW1 is stronger than that during QBO easterly phases and weaker than that during QBO westerly phase. That's the first difference. When analyzing the mechanism (including suggestions on other mechanisms provided by referee), there are also differences. It will be discussed in later comments.

[Figure]

Fig. 1. (upper panel) Amplitude of the migrating diurnal temperature tide in the MLT region at the equator. (lower panel) The zonal wind in the lower stratosphere at the equator.

El Nino is truly an important issue for QBO disruption (Barton et al., 2017) and modulating the DW1 (Kogure et al., 2021). The 2016 QBO disruption has been confirmed to have a close causal relationship with the 2015/16 extreme El Niño event (Newman et al., 2016; Osprey et al., 2016; Barton and Mccormack, 2017; Coy et al., 2017). The 2015/16 El Niño substantially weakened the subtropical easterly jet, allowing enhanced Rossby wave propagation from the extratropics into the deep tropics near 40 hPa (Barton and Mccormack, 2017). These amplified Rossby waves subsequently broke and deposited momentum near the QBO westerly core, rather than at the climatological zero-wind line, causing a pronounced deceleration. The deceleration gave rise to a persistence of westerlies at 40–15 hPa, preventing the expected transition to easterlies and ultimately leading to the QBO disruption (Newman et al., 2016; Osprey et al., 2016; Coy et al., 2017; Barton and Mccormack, 2017; Kang et al., 2022; Wang et al., 2023). The QBO disruption was accompanied by a marked strengthening of the Brewer–Dobson residual circulation, thereby intensifying tropical upwelling. This upwelling contributed to an upward displacement of westerlies in the tropical lower stratosphere (Coy et al., 2017), modifying the transport and distribution of trace gases such as water vapor. The persistent westerlies also created conducive background conditions for the vertical propagation of DW1. Nevertheless, not all strong El Niño events trigger QBO disruptions. In the 2015/16 case, the QBO westerly wind core was weaker and Rossby wave activity stronger than in other extreme events, such as the 1998 El Niño (Barton and Mccormack, 2017).
The role El Nino in triggering QBO disruption is given in line 48-61 (manuscript)

[Figure]

Fig 2. Time evolution of observed sea-surface temperatures in the NINO3.4 region from October 2014 to October 2016 according to two different analyses, OIv2 (dark blue line) and OSTIA (light blue line). The SST NINO3.4 mean seasonal cycle, defined as the climatological SST variations during the year, is represented by the red line. The difference between the red and blue lines is the SST anomaly.

[Figure]

Fig 3. ECMWF ensemble forecasts of monthly mean NINO3.4 SST anomalies with respect to the 1981 to 2010 climate issued on (c) 1 December 2015. The dotted line shows the observed evolution of SST anomalies.

Here we give the example of Fig 2. and Fig 3. (https://www.ecmwf.int/en/newsletter/151/meteorology/2015-2016-el-nino-and-beyond). During the QBO disruption event (February-April), SST NINO3.4 anomaly is in a declining trend from ~2°C – ~0.5°C. The role of El Niño should be considered influencing the DW1. The variation of heating source will be discussed in response 3. After in-depth analysis, we tend that the changes in DW1 are a compound effect of El Nino and QBO disruption.

**Comments 2:** I recommend that the introduction should review wind impact on DW1. Many papers (including cited papers already) have demonstrated that QBO wind and wind shear modulate DW1. **Response 2:** Thanks for the helpful comments. The mechanism including wind impact on DW1 will be reviewed in introduction, please see line 77-85 (manuscript). Further mechanism discussion is supplemented in discussion part. We reorganized the discussion part, the QBO wind and wind shear locate Section 4.2.

**Comments 3:** Many papers have demonstrated heating related to El Nino impacts rather than

heating related to QBO impact. The authors should cite some proper references to show that modulation of heating due to QBO is larger or comparable to that due to El Nino in introduction.

**Response 3:** Thanks for the valuable comments. In the current version of the article, we report an increase in radiative heating of water vapor and ozone, then link it to the amplification of DW1. We lack an analysis of the reasons for the increasing in water vapor or ozone. After reviewing, we found it a compound effect of QBO disruption and El Nino. The mechanism underlying is reviewed and supplemented in the manuscripts.

**$H_2O$:** In the troposphere, during the 2015/2016 El Nino (Johnston et al., 2022), fractional moisture anomalies are small in the tropical lower troposphere (~2%–3% $ONI^{-1}$) and increase with height into the middle troposphere. In the upper troposphere and lower stratosphere (UTLS), the positive signal becomes much stronger (up to 10% $ONI^{-1}$). So, the role of 2015/2016 El Nino contribute most in increasing of the water vapor concentration in UTLS.

[Figure]

Fig 1. Time–height cross sections of WACCM monthly zonal mean fractional specific humidity (%) from 2007 to 2018 within 10°N–10°S. (Johnston et al., 2022)

The occurrence of 2016 QBO disruption introduces a shear transition from westerly to easterly near 40 hPa, which strengthens tropical upwelling and lowers cold-point temperatures. This dynamical response injects $H_2O$-poor air into the lower stratosphere, partially offsetting the El Niño–driven moistening. (Diallo et al., 2018). The water vapor concentrations are still above the climatological seasonal cycle under the modulation of these two phenomena.

This part is supplemented in line 369-375 (manuscript).

**$O_3$:** Concerning the ozone, we care about the region in the upper stratosphere. $O_3$ mixing ratio decreases throughout the tropics during El Niño due to the enhanced tropical upwelling, bringing air poor in $O_3$ from the troposphere. So, the role of El Niño is damping the concentration of $O_3$. During the period of QBO disruption event, the westerly wind shear that appears between 30 and 10 hPa reduces the upward motion of the BD-circulation and causes positive $O_3$ anomaly. Therefore, most of the increase in ozone is probably due to QBO disruption. However, whether the ozone heating contribute to the DW1 should be more investigated as discussed in Comment 6 and Response 6.

**Comments 4:** The authors should evaluate the differences are statistically significant. While I believe that the QBO signal (QBOW – QBOE) is significant, as established in many previous papers, it is unclear whether the other differences meet statistical significance. I recommend

showing the standard error of the mean values instead of the standard deviation. The methodology is explained in Kogure et al. (2023).

**Response 4:** Thanks for the valuable comments. In Kogure et al. (2023), The data used by the author for significance testing were one-to-one (7 months to 7 months), while the data in this article were sometimes one-to-many (3 months to 18 months). When calculating standard error using equation SD/ √ (sample size), the sample size cannot be confirmed. So here we apply Welch's t-test. The results are given in the form of dots or weighted lines in the figures. We give the example of DW1 difference in MLT and its significance test result (Figure 2 in manuscript and its dotted region).

[Figure]

**Comments 5:** I suggest decomposing the DW1 components in the WACCM-X into Hough modes to better identify whether propagating or trapped modes are being enhanced, especially in the stratosphere. If you have a MATLAB license, you can use the code in Wang et al. (2016). If not, you can get Python code in the following git hub repository.

In the current analysis, the discussion above 60 km seems fine because the signal of (1,1) mode can be seen clearly. However, below 60 km, contamination from trapped modes seems significant, potentially introducing errors.

**Response 5:** Thanks for the valuable comments. We really appreciated the referee for providing the Hough modes code. We apply the program from https://github.com/masaru-kogure/Hough_Function.

[Figure]

Fig 1. Amplitude profiles of DW1a) (1,1) and b) (1, -2) modes during different QBO phases

The trapped mode dominant below 60 km. As shown in Fig 1, The amplitude of (1, -2) mode is larger in this region. However, there is less difference between different QBO phases. In (1, 1) mode, there is clear difference between QBO easterly and westerly phases. we give an example of DW1 equatorial amplitude and (1, 1) mode amplitude at 95 km. There is less difference between DW1 and its (1, 1) mode. We can infer that (1,1) mode reveals almost all QBO variability in MLT region. This may be because the QBO dominant only at equator. For all the structure of Hough mode, the (1, 1) mode fit this latitude structure mostly.

[Figure]

Fig 2. The amplitude time series of equatorial DW1 and (1, 1) Hough mode at 95 km.
This part is supplemented in line 300-306 (manuscript).

**Comments 6:** I wonder whether ozone heating can significantly modulate propagating DW1 tide, since the vertical thickness of ozone heating (~30 km) is too large to generate DW1 (1,1) mode efficiently (Chapman and Lindzen, 1970). In addition, Hagan (1996) said, "However, because the diurnal component excited by UV heating the stratosphere and mesosphere is out of phase with the dominant component, the former acts to suppress the latter."

To justify the claim that the ozone heating strengthens the (1,1) mode, I strongly recommend computing a ratio of the tidal amplitude between 2016 and QBOE. If the authors are correct, the

ration (2016/QBOE) should increase above 30 km with height, but the increase should not seem below 30 km.

Noth that if ozone heating does contribute to the DW1 (1,1) mode generation, the phase should change in the stratosphere as well. Considering the combination of trigonometric functions, the phase should change significantly in the stratosphere when additional source exists there. For more on this, see section 4.2 in Kogure et al. (2023).

**Response 6:** Thanks for the valuable comments. We review the papers that ozone heating that influencing the (1, 1) mode. There are many papers talking about relationship between the thickness of ozone layer and DW1. Since the vertical thickness of ozone heating (~40 km) is too large while the DW1 wavelength is relatively short. The efficiency of generating DW1 (1,1) mode will not be large (e.g., Butler et al., 1962; Chapman and Lindzen, 1970; Hagan, 1999; Garcia, 2023). While using the GSWM, Tide Mean Assimilation Technique (TAMT) modeling DW1 variation (Hagan, 1996; Ortland et al., 2017), the simulation result show that DW1 generated by ozone heating will be out of phase with the DW1 generated by water vapor heating, resulting the reduced amplitudes. In the MLS observations (Wu et al., 1998), the tropical diurnal tide at 1.0 hPa (~49.5 km) is largely depressed as shown in Fig 1. Wu et al. (1998) suggested that the interference between the vertically propagating (1,1) tide and the component from local ozone heating that contributes significantly to the propagation and generation of the diurnal tide.

[Figure]

Fig 1. Diurnal temperature amplitudes observed by MLS at pressure levels of 22-0.46 hPa and latitudes of 25°S-25°N. The diurnal tide is extracted from the A-D temperature differences using a least square fitting method and contoured at an interval of 0.4 K.

Indeed, the view that ozone has a positive effect on DW1 does need to be revised. In our result (Fig 2.), the (1, 1) mode revealed by SABER supported the result from MLS observations (Wu et al., 1998). We found suppressed amplitudes around 50 km. The (1, 1) mode ozone heating rate also reach peaks a bit below 50 km. However, whether the ozone heating modulated DW1 (1, 1) mode, there needs more detailed investigation like model simulation from Kogure (2023).

[Figure]

Fig 2. The temperature (solid lines) and ozone heating (dashed lines) DW1 (1, 1) mode amplitude profile.
This part is given in line 408-420 (manuscript).

**Comments 7:** I feel that the authors should discuss the relation between tropospheric water vapor, stratospheric ozone and the QBO disruption. The authors should cite some proper references that the QBO disruption enhanced the water vapor or ozone, or propose some plausible mechanisms of the enhancement.
I personally speculate QBO might modulate the upper tropospheric water vapor and the stratospheric ozone, but I am not sure the modulation enhances or suppress them.
**Response 7:** Thanks for the helpful comments. This part has been discussed in response 3.

**Comments 8:** I guess that the authors showed the QBO impact on the tidal vertical wavelengths to exclude an impact of Doppler-Shift due to the QBO wind on tide. It should be noted that QBO wind does not influence their vertical wavelengths above ~40 km altitude. However, I personally doubt the result regarding the vertical wavelengths because they exclude the contamination from trapped modes. Also, the QBO modulates the wind shear around 18N/S (McLandress, 2002; Mayr and Mengel, 2005; Sakazaki et al., 2013), affecting the (1,1) mode. The authors do not discuss on this effect at all. Although the physical meaning of this shear effect is debated, it undeniably influences the (1,1) mode.
**Response 8:** Thanks for the valuable comments. We have extracted DW1 (1, 1) by adopting the Hough mode decomposition (from Kogure's program) to avoid the contamination from trapped modes. Here we gave the result of the phases structure of DW1 (1, 1) mode. The wavelengths

statistic is also updated. The significance test method is bases on the latter comment.

[Figure]

From SABER's result, there is a clear phase shift between the QBO westerly phases and easterly phases at around 40 km. During the QBO disruption events, the phase structure is similar to that during the QBO westerly phases. From this result, the wavelengths during QBO disruption seem shorter than that during QBO easterly phases at ~15 km - ~ 40km and longer than that at ~40 - ~75 km. From ~75- ~ 105 km, the wavelengths show less difference between different QBO phases. In WACCM-X result, there is less difference between different QBO phases. The wavelengths statistic result show similar feature.

Table 1. The comparison of mean (left of the slash) and standard deviations (right of the slash) of DW1 (1, 1) wavelengths (in km) revealed by SD-WACCM-X and SABER from stratosphere to MLT between QBO westerly phase, easterly phase, 2016 disruption event and 2020 disruption event calculated from February to April.

| Data | SD-WACCM-X | | | SABER | | |
|---|---|---|---|---|---|---|
| altitude | ~15 km – ~ 40 km | ~40 km – ~ 75 km | ~75 km – ~105 km | ~15 km – ~ 40 km | ~40 km – ~ 75 km | ~75 km – ~105 km |
| Westerly | 22.97/1.49 | 34.47/1.79 | 25.10/1.84 | 21.81/1.44 | 33.12/1.78 | 21.29/1.04 |
| Easterly | 22.51/1.73 | 34.42/2.15 | 25.60/2.20 | 24.46/1.99 | 30.84/2.35 | 20.56/1.30 |
| 2016 | 22.56/1/33 | 33.26/1.58 | 25.58/2.03 | 21.48/2.31 | 33.32/2.10 | 21.28/0.85 |

| 2020 | 22.71/1.87 | 33.80/2.68 | 26.27/2.41 | 21.08/1.77 | 34.24/1.46 | 20.39/1.35 |

This analysis is given in section 3.2 (manuscript).

The wind shear structure around 18°N/S is given below and will be supplemented in the manuscripts. We still applied the QBO phase classification and statistics method used in the manuscript to analyze. During the 2016 disruption event, the $|\partial u/\partial y|$ at 18N show similar structure to the QBO westerly phases from 25 to 35 km. From 35 km to 45 km, large positive values are found, which is unique from other QBO phases. This is one mechanism difference. The $|\partial u/\partial y|$ at 18S show similar structure to that at 18N (no peak at 65 km), but the amplitude is smaller. According to the theory of McLandress (2002) and Mayr and Mengel (2005), large wind shear around 18N/S reduces the DW1 amplitude. Hence, wind shear seems amplify the DW1 amplitude from 25 to 35 km and weaken the DW1 amplitude from 35 to 45 km.

[Figure]

This discussion of wind shear is given in line 452-370 (manuscript).

**Comments 9:** Based on my knowledge, El Nino primarily influences the water vapor in the troposphere rather than QBO. Indeed, the disruption was most likely triggered by a strong El Nino (Coy et al., 2017; Newman et al., 2016; Osprey et al., 2016). How do the authors exclude the impact of El Nino on the DW1 tides?

**Response 9:** Thanks for the valuable comments. We need to revise our article for lacking an analysis of the reasons for the increasing in water vapor. The increasing water vapor is due to the compound effect from El Nino and QBO disruption. The El Nino moisten the upper troposphere and lower stratosphere (UTLS) while QBO disruption reverse this trend. We have supplemented the underlying mechanism in the revised version which is discussed in response 3.

**Comments 10:** As for Figure 6, I recommend showing the absolute values of the heating rather than

the relative (percent) values. The rate sometimes emphasizes the difference too much in regions with small absolute values. In addition, I suggest showing the heating values integrated, averaged, or smoothed in vertical. Since the propagating DW1 has an ~20–30 km vertical wavelength, a effective thickness of heating should be 10–15 km (Chapman and Lindzen, 1970).

**Response 10:** Thanks for the valuable comments. We reorganized the Figure 6 and give the comparison between the smoothing result and original result. Also, the (1, 1) mode is given in Fig. 3.

**Original:**

[Figure]

Fig 1. Latitude-altitude distribution of the differences between different QBO phases of (a-e) water vapor heating rate DW1 component from MERRA-2, (f-j) ozone heating rate DW1 component from SABER and (k-o) longwave heating rate from SD-WACCM-X.

**Smoothing:**

[Figure]

Fig 2. Same as Fig 1. but smoothing in vertical.

Here we apply 10 km window smoothing. Comparing Fig. 1 and Fig. 2, the ozone and long wave heating rate show nearly the same feature. However, the water vapor heating show its unique feature. There is a large decrease (Changes in order of magnitude) at around 12 km. This is also shown in (1, 1) mode of heating sources (Fig. 3c). When applying smoothing, the peaks will move downward during to the weak amplitude above 12 km (Comparing Fig. 1a-e & 2a-e). This will introduce the error. We want to show the vertical structure so we tend to use the original difference result.

[Figure]

Fig. 3 Heating rate profiles of the DW1 (1, 1) mode between different QBO phases and their differences. (a, b) give the water vapor heating profile and its difference derived from MERRA2. The bold lines indicate the difference that are significant at the 95% confidence level.

As shown in Fig. 3b, during the 2016 QBO disruption event, the largest difference occurs at 10.5 km, which is 0.043 K/day. The percentage change relative to QBOE is about 8%. This is another mechanism difference (difference in heating sources). As in manuscripts, the DW1 amplitude vary about 20.5%. The water vapor heating could only explain 39% of the amplitude difference.

This discussion of water vapor (1, 1) mode is given in line 425-432 (manuscript).

Comments 11: H-Liu (2010; 2018) should be cited.
Liu, H.-L., Bardeen, C. G., Foster, B. T., Lauritzen, P., Liu, J., Lu, G., ⋯ Wang, W. (2018). Development and validation of the Whole Atmosphere Community Climate Model with thermosphere and ionosphere extension (WACCM-X 2.0). Journal of Advances in Modeling Earth Systems, 10, 381–402. https://doi.org/10.1002/2017MS001232
Liu, H.-L., et al. (2010), Thermosphere extension of the Whole Atmosphere Community Climate Model, J. Geophys. Res., 115, A12302, doi:10.1029/2010JA015586.
**Response 11:** We have revised.

**Comments 12:** ".etc"-> "and so on."
**Response 12:** We have revised.

**Comments 13:** Clarify a full width at half maximum.
**Response 13:** We have revised. Please see line 197-198 (manuscript).

**Comments 14:** "*Within each QBO cycle, the DW1 amplitude in the stratosphere below 40 km leads that in the MLT region by one to two months.*"
I am not sure why the DW1 in the stratosphere lead that in the MLT because the DW1 tide does not take one month to propagate from the stratosphere to the MLT. I guess that this discrepancy could attributed to the trapped mode variation in the stratosphere.
**Response 14:** We retrieve the DW1 (1, 1) mode from the original DW1 data.

[Figure]

There do exist a lead. it is interesting result. However it is out of the scope of our aims, so we delete it in the manuscript.

**Comments 15:** I assume that the authors calculated the standard deviation from the phase itself. However, I think the phase does not have a symmetric distribution. Also, the values change cyclically (e.g., it jumps from 4pi to -4pi), causing the overestimation of the standard deviation. Indeed, the standard deviation is very larger in larger than 2 while it is very small around 0. In such a case, I recommend the following steps. First, you calculate averages and standard devotion (or error) of sine and cosine Fourier components, and then you calculate the average phase and its confidential interval using the error propagation.
Note that if the authors calculate the average from the phase itself, it must distort the vertical wavelengths. For example, 4 pi and -3.9pi are almost the same phase, but the average value is almost 0.
**Response 15:** Thanks for the valuable comments. This is really a good method. we have applied it. The result is given in response 8 and the introduction is given in line 318-321 (manuscript).

**Comments 16:** "So," -> "; hence,"
**Response 16:** We have revised.

---

## Author Comment (AC2)

In this revised version, three major improvements are introduced:

(1) The relationship between the 2015/16 El Niño and the 2016 QBO disruption is reviewed, with emphasis on how the El Niño event triggered the QBO disruption and how both phenomena jointly influenced the DW1 heating sources, particularly water vapor.

(2) The (1,1) mode of the DW1 is extracted and analyzed, and its amplitude and phase responses to the event are presented in Sections 3.1 and 3.2.

(3) The discussion of mechanisms is expanded. Section 4 is reorganized into three subsections addressing tidal heating, tidal propagation, and tidal–gravity wave interaction. The result of (1, 1) mode water vapor heating is given in Section 4.1. The role of ozone heating is discussed in Section 4.1. The contribution of zonal wind latitudinal shear and gravity wave drag are also discussed in 4.2 and 4.3, respectively.

**Major Comments:**

1. The authors attribute all of the variability in the DW1 in 2016 and 2020 to be related to the QBO disruptions that occurred during this time period. They thus neglect other possible sources of variability. This is especially pertinent for the 2016 time period, when there was a strong ENSO, which has also been shown to influence the DW1 in previous studies. Given that the DW1 anomalies are significantly stronger in the 2016 case compared to the 2020 case, there could be additional contribution from the ENSO event during 2016. The DW1 anomalies in 2016 may thus not be solely due to the QBO disruption. The authors should consider the potential role of other sources of variability in the DW1 and how these may influence the results.

**Response 1:** Thanks for the valuable comments. In the current version of the article, we report an increase in radiative heating of water vapor and ozone, then link it to the amplification of DW1. We lack an analysis of the reasons for the increasing in water vapor or ozone.

The 2016 QBO disruption has been confirmed to have a close causal relationship with the 2015/16 extreme El Niño event (Newman et al., 2016; Osprey et al., 2016; Barton and Mccormack, 2017; Coy et al., 2017). The 2015/16 El Niño substantially weakened the subtropical easterly jet, allowing enhanced Rossby wave propagation from the extratropics into the deep tropics near 40 hPa (Barton and Mccormack, 2017). These amplified Rossby waves subsequently broke and deposited momentum near the QBO westerly core, rather than at the climatological zero-wind line, causing a pronounced deceleration. The deceleration gave rise to a persistence of westerlies at 40–15 hPa, preventing the expected transition to easterlies and ultimately leading to the QBO disruption (Newman et al., 2016; Osprey et al., 2016; Coy et al., 2017; Barton and Mccormack, 2017; Kang et al., 2022; Wang et al., 2023). The QBO disruption was accompanied by a marked strengthening of the Brewer–Dobson residual circulation, thereby intensifying tropical upwelling. This upwelling contributed to an upward displacement of westerlies in the tropical lower stratosphere (Coy et al., 2017), modifying the transport and distribution of trace gases such as water vapor. The persistent westerlies also created conducive background conditions for the vertical propagation of DW1. Nevertheless, not all strong El Niño events trigger QBO disruptions. In the 2015/16 case, the QBO westerly wind core was weaker and Rossby wave activity stronger than in other extreme events, such as the 1998 El Niño (Barton and Mccormack, 2017).

For the role of modulating heating source of DW1, after reviewing, we found it a compound effect of 2016 QBO disruption and 2015/16 El Nino event.

In the troposphere, during the 2015/2016 El Nino (Johnston et al., 2022), fractional moisture

anomalies are small in the tropical lower troposphere (~2%–3% ONI$^{-1}$) and increase with height into the middle troposphere. In the upper troposphere and lower stratosphere (UTLS), the positive signal becomes much stronger (up to 10% ONI$^{-1}$). So, the role of 2015/2016 El Nino contribute most in increasing of the water vapor concentration in UTLS.

[Figure]

Fig 1. Time–height cross sections of WACCM monthly zonal mean fractional specific humidity (%) from 2007 to 2018 within 10°N–10°S. (Johnston et al., 2022)

The occurrence of 2016 QBO disruption introduces a shear transition from westerly to easterly near 40 hPa, which strengthens tropical upwelling and lowers cold-point temperatures. This dynamical response injects $H_2O$-poor air into the lower stratosphere, partially offsetting the El Niño–driven moistening. (Diallo et al., 2018). The water vapor concentrations are still above the climatological seasonal cycle under the modulation of these two phenomena.

How the El Niño event triggered the QBO disruption is supplemented in line 48-61 (manuscript).

How both phenomena jointly influenced the DW1 heating sources, particularly water vapor is supplemented in line 369-375 (manuscript).

2. The authors focus on analysis of heating rates to explain the reason for the anomalous DW1 amplitudes during the 2016 and 2020 QBO disruptions. However, previous studies, such as Mayr and Mengel (2005, doi:10.1029/2004JD005055) and Wang et al. (2024, dot:10.5194/acp-24-13299-2024), illustrate the potential importance of tide-gravity wave interactions in the DW1 variability due to the QBO. The manuscript provides no details about the possible influence of tide-gravity wave interactions on the DW1 anomalies during the 2016 and 2020 QBO disruptions. Given these previous studies, the possible impact of tide-gravity wave interactions should also be considered.

**Response 2:** Thanks for the valuable comments. We miss the effect of tide-gravity wave. In line 83 – 85, we cite some papers to draw out the role of the gravity wave. In discussion part, the analysis of gravity wave (GW) is supplemented. Please see Section 4.3.

**Gravity wave drag, SD-WACCM-X**

[Figure]

Here we give a brief discussion. The latitude-height distribution of gravity wave drag on DW1 in different QBO phases are shown above. We could see that above 105 km GW tend to damp the DW1 amplitude at nearly all latitude. Below ~105 km, the gravity wave tends to damp the DW1 amplitude at equator and strengthen the DW1 amplitude at subtropical. There are differences in the amplitude of gravity wave drag between different QBO phases. We give the drag amplitude differences between different QBO phases. The dots represent the region over 95% statistical significance.

**Gravity wave drag, SD-WACCM-X**

[Figure]

As shown in Figure c, during the QBO westerly phases (QBOW), the damping at equator and strengthening at subtropical is stronger than QBO easterly phases (QBOE). When comparing the 2016 QBO disruption and QBO easterly phases (Figure d), the pattern is similar to the QBO westerly minus QBO easterly. While comparing the 2020 QBO disruption and QBO easterly phases (Figure e), the difference is weaker compared to 2016-QBOE.

3. The ozone heating rates are derived from TIMED/SABER observations. Only a brief statement (lines 99-100) is included for the calculation of the ozone heating rates. Additional details and appropriate references should be included for the calculation of the ozone heating rates.

**Response 3:** Thanks for the valuable comments. We apply the Strobel/Zhu model (Strobel, 1978; Zhu, 1994) to calculate the heating rate of Hartley band, Huggins band and Chappuis band. The calculation details will be supplemented in Appendix B and section 2.6.

4. The description of the SD-WACCM-X model in Section 2.2 needs to be revised as it is not completely correct. The manuscript states that WACCM-X consists of two parts, WACCM and the TIE-GCM. This is incorrect as WACCM-X is a single model that extends WACCM into the upper thermosphere and includes additional ionosphere and thermosphere processes. These additional processes are largely based on the TIE-GCM,

but it is incorrect to state that the entire upper atmosphere of WACCM-X is the TIE-GCM as there are some differences between the two models.

**Response 4:** Thanks for the valuable comments. The current manuscript's description of SD-WACCM-X as "consisting of two parts, WACCM and the TIE-GCM" is indeed imprecise. WACCM-X is a single, self-consistent whole-atmosphere model that extends WACCM from the surface to the upper thermosphere/ionosphere (~500–700 km). While many of the thermosphere–ionosphere physics modules (e.g., electrodynamo, $O^+$ transport, electron/ion energetics) were adapted from the NCAR TIE-GCM, they have been re-implemented within the WACCM-X dynamical core and coupled to WACCM's lower and middle atmosphere through a dedicated interface layer. Consequently, WACCM-X is not simply WACCM plus TIE-GCM; rather, it is a unified model in which the upper-atmosphere processes are integrated into the WACCM framework, with some differences in numerical methods, boundary conditions, and physical parameterizations. We will revise Section 2.2 to make this point explicit. Please see line 131-137 (manuscript).

5. The data availability section should include the availability of the SD-WACCM-X output.

**Response 5:** We have given it in data availability part.

Minor Comments:
1. Line 15: "in mesosphere and" should be "in the mesosphere and"

**Response 1:** We have revised.

2. Lines 47-48: This sentence should be revised as it is unclear what is meant by "upward westerly wind" and "upward easterly wind".

[Figure]

**Response 2:** Thanks for the valuable comments. The normal QBO phases manifests as alternating downward-propagating westerly and easterly winds (red arrow and blue arrow). During the period of disruption event, unique westerly wind occurs and then propagate upward. We want to distinguish between normal downward-propagating and unique upward-propagating, but our sentences are too simple and obscure the meaning.

3. Line 64: "SOBO disruption" should be "QBO disruption"

**Response 3:** We have revised.

4. Line 90: "sun-synchronal" should be "sun-synchronous"

**Response 4:** We have revised.

---

## Author Response (AR2)

In this revised version, several improvements are introduced:

(1)  The logic of the Discussion section has been organized. A summary paragraph is added to give the logic of mechanism analysis. The subtitle is changed to be more detailed. Then, the transitional sentences between each section are added.

(2)  The grammar of the entire manuscript has been checked and corrected.

(3)  A comparison between disruption events and QBOW is supplemented, including the Abstract, Result section, and Summary section.

(4)  The relationship between QBO and gravity wave drag has been supplemented.

(5)  Possible reasons for differences in wavelength statistics compared to previous studies is discussed.

(6)  The discussion of tidal dissipation using amplitude ratio method is added.

(7)  The connection between water vapor radiative heating variations and the joint modulation by ENSO and QBO disruption has been strengthened.

**Response to Reviewer 1:**

We would like to express our sincere appreciation to the reviewers for their comprehensive assessment and insightful comments, which have been so helpful in enhancing the substance of our work.

Comment 1: The discussion section has become somewhat disorganized, making it difficult to discern the central arguments of the paper. I recommend that the authors reorganize the narrative, clarify the focus, and streamline the discussion and conclusion based on the updated results.

Response 1: Thanks for the helpful comments. In discussion section, our aim is to study how the QBO disruptions and 2015/16 extreme El Nino modulate the DW1 by several mechanisms from the lower atmosphere to upper atmosphere. Therefore, the discussion part is organized as tidal heating (troposphere and stratosphere), tidal propagation (from stratosphere to mesosphere) and tide-gravity wave interaction (significantly in mesosphere and lower thermosphere, as shown in Figure 11). However, these subtitles are so broad that they obscure what we want to express. Consequently, at the beginning of the Section 4, a summary paragraph is added to give the logic of mechanism analysis. Furthermore, the subtitle will be changed to be more detailed. Then, the transitional sentences between each section are added.

Comment 2: Additionally, while I am not a native English speaker, I strongly recommend that the manuscript be proofread by a native speaker. Some sentences are structurally awkward or grammatically incorrect.

Response 2: Thanks for the helpful comments. We have carefully reviewed our manuscript and corrected faults.

Comment 3: The title refers to "Migrating diurnal tide anomalies," but it is unclear what is meant by "anomalies." Based on the results, the DW1 tide appears comparable to that during a typical QBOW phase, which raises doubts about whether "anomaly" is the appropriate term. Moreover, the

manuscript discusses differences from QBOE but provides minimal comparison with QBOW. The authors should clarify: Whether the 2016 and 2020 events are similar to a typical QBOW period, and if so, explain why they resemble QBOW conditions.

If these events are instead unique and distinct from QBOW, the manuscript should discuss in detail how they differ.

In short, the key points of the paper are not clearly conveyed.

**Response 3:** Thanks for the helpful comments. The comment that the reviewer give is reasonable. In our studies, the comparison with QBOW and QBOE are both presented. However, in the previous version of the manuscript, we mainly present the differences between events and QBOEs in Abstract and Summary. That's because, as the previous response (round 1) mentioned, if there are no events, the QBO should evolve into QBOE. That's why we focus on the comparison between QBOE and event in the previous version of the manuscript. However, as the reviewers pointed out, the differences of DW1 between events and both QBOW and QBOE should be clearly presented.

The differences are summarized as follows. In term of the amplitude difference:

(1) In MLT region, during the 2016 event, the amplitude at equator is smaller than that during QBOW about 10.2% and larger than that during QBOE about 20.5%. In subtropical latitude (30°N/S), the amplitude is larger than that during QBOW about 4.6% and larger than that during QBOE about 14.4%. During the 2020 event, the amplitude in the equatorial MLT is smaller than that during QBOW about 21.1% and larger than that during QBOE about 6.0%. In subtropical latitude (30°N/S), the amplitude in the equatorial MLT is smaller than that during QBOW about 21.1% and larger than that during QBOE about 6.0%.

(2) In the upper stratosphere, during the 2016 event, the amplitude of DW1 (1, 1) mode is smaller than that during QBOW about 10.9% and larger than that during QBOE about 21.1%. During the 2020 event, the amplitude of DW1 (1, 1) mode is smaller than that during QBOW about 12.5% and larger than that during QBOE about 17.0%

In term of the mechanism difference:

(3) During the 2016 event, the water vapor radiative heating is enhanced by 8.3% relative to QBOE and 10.9% relative to QBOW due to the compound effect of QBO disruption event and 2015/16 El Nino event.

(4) Comparing the tide-gravity wave interaction during different QBO phases and QBO disruption, the GW forcing during 2016 event is stronger than that during the QBOW and QBOE.

These summary of difference have been supplemented in Abstract and Summary.

Comment 4: Additionally, the abstract attributes the tidal enhancement during the 2016 disruption to zonal wind shear and gravity wave drag. While the shear is clearly related to the QBO, it is unclear whether gravity wave drag is also linked to the QBO. It may be more related to the SAO or gravity wave source activity. Please expand on the relationship between QBO and gravity wave drag.

**Response 4:** Thanks for the helpful comments. This question has been discussed in Wang et al., (2024). QBO-dependent zonal wind shear and associated zero-wind lines filter the upward-propagating gravity waves that can reach the mesosphere, making the gravity wave drag exhibit QBO-like feature. In the tropical region of mesosphere, because of the strong interaction between the GWs and the semi-annual oscillation (SAO) in zonal wind, the GWs in mesosphere exhibit a weak QBO signature. QBO-related variations in GWs primarily exists in the mid-latitude

mesosphere.

Wang, J., Li, N., Yi, W., Xue, X., Reid, I. M., Wu, J., Ye, H., Li, J., Ding, Z., Chen, J., Li, G., Tian, Y., Chang, B., Wu, J., and Zhao, L.: The impact of quasi-biennial oscillation (QBO) disruptions on diurnal tides over the low- and mid-latitude mesosphere and lower thermosphere (MLT) region observed by a meteor radar chain, Atmos. Chem. Phys., 24, 13299-13315, 10.5194/acp-24-13299-2024, 2024.

This part could be seen in Line 554-558 (in marked-up manuscript) or Line 525-530 (in all-accepted manuscript).

Comment 5: According to my understanding, westerly winds increase DW1 tidal vertical wavelengths, leading to amplified tidal amplitudes (Forbes and Vincent, 1989; Kogure and Liu, 2021). This is inconsistent with the findings presented in this manuscript. I recommend computing theoretical vertical wavelengths at the equator using Equation (14) from Forbes and Vincent (1989). It is important to note that vertical wavelengths in the mesosphere are influenced not by the QBO but by winds in the mesosphere. If the authors intend to discuss vertical wavelength and dissipation processes in the mesosphere, they should include wind profiles from that region.

**Response 5:** Thanks for the helpful comment. The statistical results derived from the method used in this study were compared with those obtained from the approach proposed by Kogure and Liu (2021, hereafter KL21). The comparison reveals that, in the lower stratosphere of the SABER data, the standard deviation of the wavelengths estimated by the KL21 method is sufficiently large to obscure distinctions between different QBO phases. For instance, as shown in Table 2, the mean wavelength difference between QBOW and QBOE in the lower stratosphere (~18–32 km) is only 0.62 km, whereas the corresponding standard deviation reaches approximately 3 km. After careful consideration, we decided to preserve our results.

According to the theoretical framework proposed by Forbes and Vincent (1989) and Kogure and Liu (2021), zonal winds modify the intrinsic frequency of tides through Doppler shifting, thereby altering their vertical wavelengths. Specifically, westerly winds lead to longer DW1 vertical wavelengths, whereas easterly winds result in shorter ones. However, the result of wavelengths shown in Table 1 differs from that reported in previous studies. This discrepancy can be attributed to differences in methodology. In this study, the vertical wavelengths are determined from the phase difference between adjacent peaks ($+\pi$). In the stratosphere, one of these peaks typically occurs in the lower stratosphere (~18 km) and the other in the upper stratosphere (~40 km). Consequently, the estimated wavelengths encompass the combined influences of both the QBO and SAO, producing a mixed result that deviates from earlier findings. The advantage of our method is that it only requires two points to calculate the wavelength. The result is as following:

Table 1. Wavelengths statistic using peak difference method

| Data | SD-WACCM-X | | | SABER | | |
|---|---|---|---|---|---|---|
| altitude | ~15 km – ~ 40 km | ~40 km – ~ 75 km | ~75 km – ~105 km | ~15 km – ~ 40 km | ~40 km – ~ 75 km | ~75 km – ~105 km |

| | | | | | | |
|---|---|---|---|---|---|---|
| Westerly | 22.97/1.49 | 34.47/1.79 | 25.10/1.84 | 21.81/1.44 | 33.12/1.78 | 21.29/1.04 |
| Easterly | 22.51/1.73 | 34.42/2.15 | 25.60/2.20 | 24.46/1.99 | 30.84/2.35 | 20.56/1.30 |
| 2016 | 22.56/1/33 | 33.26/1.58 | 25.58/2.03 | 21.48/2.31 | 33.32/2.10 | 21.28/0.85 |
| 2020 | 22.71/1.87 | 33.80/2.68 | 26.27/2.41 | 21.08/1.77 | 34.24/1.46 | 20.39/1.35 |

In our result, the standard deviation does not obscure wavelength differences during different QBO phases.

The wavelength is then calculated using the KL21 method. The method is as follows. The tidal vertical wavenumbers are firstly derived from the (1,1) mode tidal phase with the least-square method in a range of 2 scale heights (∼14 km) by 0.2 scale height steps. The vertical wavenumbers were then averaged in three regions: the QBO zonal wind height range (18–32 km), Stratospheric Semi-Annual Oscillation zonal wind height range (32 – 60 km), and Mesosphere and lower thermosphere Semi-Annual Oscillation zonal wind height range (60 – 100 km). Then the wavelengths are calculated by $2\pi/wavenumber$. The result is as following:

Table 2. Wavelengths statistic using KL21 method

| Data | SD-WACCM-X | | | SABER | | |
|---|---|---|---|---|---|---|
| altitude | ~18 km – ~ 32 km | ~32 km – ~ 60 km | ~60 km – ~100 km | ~18 km – ~ 32 km | ~32 km – ~ 60 km | ~60 km – ~100 km |
| Westerly | 22.36/1.09 | 26.57/1.16 | 25.37/1.35 | 24.59/2.81 | 25.48/1.57 | 21.10/0.44 |
| Easterly | 22.01/1.27 | 26.57/1.33 | 25.54/1.39 | 23.97/3.23 | 23.91/2.48 | 20.52/0.53 |
| 2016 | 21.96/0.77 | 26.23/0.97 | 25.48/1.43 | 24.49/2.55 | 24.71/1.16 | 20.93/0.23 |
| 2020 | 21.98/1.10 | 26.93/1.22 | 25.82/1.38 | 26.74/3.78 | 23.28/1.58 | 20.26/0.46 |

This result is consistent with previous studies. However, the standard deviation shown in SABER observations is so large in stratosphere that the wavelengths difference between different QBO phases is negligible to the standard deviation. There's no doubt that KL21 method is very suitable for analyzing the regional wavelengths, especially in model simulation results. However, considering the SABER observations, we retain our results.

Furthermore, in section 3.2, we tent to present the wavelengths feature during the QBO disruptions and their difference to the normal. We plan to discuss the dissipation process using the ratio method provided by the reviewer, which will locate at Section 4.2.

Possible reasons for differences in wavelength statistics compared to previous studies could be seen in Line 363-370 (in marked-up manuscript) or Line 358-365 (in all-accepted manuscript).

Comment 6: I recommend using the ratio of DW1 tidal amplitudes, rather than the absolute

difference, to account for the decrease in atmospheric density with altitude. Here's why the ratio method is advantageous: For example, comparing 2016 to QBOE using the ratio 2016/QBOE: If the ratio is larger than 1 from the lower stratosphere to the mesosphere-lower thermosphere (MLT), this suggests stronger tidal source activity in 2016 than in QBOE.

If the ratio is almost 1 in the lower stratosphere but decreases in the middle stratosphere, this implies enhanced tidal dissipation in the middle stratosphere, possibly linked to QBO.

If the ratio is almost 1 below the middle stratosphere but decreases in the upper stratosphere, the change may not be due to QBO but rather to SAO effects.

Simply say, the amplitude ratio changes, suggsting tidal dissipation varies at the altitude. This framework originates from Forbes and Vincent (1989), which I strongly encourage the authors to review carefully.

Currently, Liu et al. (2025) do not clearly identify the altitudes at which tidal dissipation increases or decreases. If the ratio does not vary in QBO-sensitive altitude ranges, it would suggest that QBO has limited influence on tidal variation.

**Response 6:** Thanks for the helpful comments. We believe the reviewer refers to Figure 4 in section 3.1 of manuscript. In that figure, we would like to analyze how the DW1 (1, 1) amplitude vary during the QBO disruption event and its difference to the normal QBO phases. It will be kept in that section. The amplitude ratio will be discussed in section 4.2 (about tidal propagation). The analysis of the method is given following.

As in Forbes and Vincent (1989), the amplitude growth equation is:

$$\frac{A(z)}{A(70)} = \exp\left\{\int_{70}^{z}\left[-k_i + \frac{1}{2H}\right]dz\right\} \tag{1}$$

where A is the amplitude, $H$ is the local scale height and $k_i$ is the imaginary part of the complex vertical wavenumber that governs damping of the amplitude profile.

When applying ratio of equation (1) like 2016 divided by QBOE, the equation become:

$$\frac{A_{2016}(z)}{A_{QBOE}(z)} = \exp\left\{\int_{70}^{z} -\left(k_{i,2016}(z) - k_{i,QBOE}(z)\right)dz\right\} \tag{3}$$

The scale height term is removed, leaving the dissipation term. Thus, the amplitude ratio changes may reflect tidal dissipation variations at the altitude.

[Figure]

Figure 1. the vertical profile of DW1 (1, 1) mode amplitude ratio derived from 2016/westerly (dark

red), 2020/westerly (dark purple), westerly/easterly (yellow), 2016/easterly (green), and 2020/easterly (orange). The dashed blue lines represent the ratio of 1.

Figure 1 presents the ratio results derived from SABER observations and SD-WACCM-X simulations. In the SABER data (Figure 1a), during the 2016 event, two distinct peaks appear in the lower stratosphere near 22 km and 30 km when comparing the disruption events with the QBOE phase (green lines), possibly indicating relatively less dissipation during the 2016 QBO disruptions. During the 2020 event, the lower peak (~22 km) is close to that during 2016 event, while the upper peak (~30 km) is relative weak to the 2016 event. This may suggest a relatively large dissipation at those heights. The SD-WACCM-X simulations reproduce a similar pattern, although the peak altitudes differ slightly. All simulated ratios remain above 1, which may indicate stronger tidal source activity from the SD-WACCM-X perspective. Overall, these results suggest that during QBO disruptions, zonal wind may result in relative less dissipation processes, thereby affecting DW1 amplitudes.

The discussion of dissipation could be seen in Line 505-525 (in marked-up manuscript) or Line 482-502 (in all-accepted manuscript).

**Response to Reviewer 2:**

We would like to express our sincere appreciation to the reviewers for their comprehensive assessment and insightful comments, which have been so helpful in enhancing the substance of our work.

1. The authors have included discussion of the 2015/2016 El Nino in relation to the 2016 QBO disruption, though they primarily discuss how the ENSO played a role in causing the QBO disruption. An important additional point is that a substantial fraction of the increased water vapor heating during the 2016 QBO disruption is likely to be due to the El Nino event that occurred during this time. This should be explicitly mentioned in both the abstract and conclusions as well as the discussion (lines 482-489).
**Response 1:** Thanks for the helpful comments. This suggestion is particularly valuable because the water vapor radiative heating during the 2016 QBO disruption event was strongly modulated by both the disruption itself and the 2015/16 extreme El Niño event. We have added this point to the Abstract, Discussion, and Summary. In particular, in the Discussion section, the paragraph describing how the ENSO and QBO disruption modulate water vapor concentration has been moved above the paragraph on water vapor heating rate, allowing a clearer connection between variations in water vapor concentration and the corresponding heating rate.
The supplement in Discussion could be seen in Line 454-466 (in marked-up manuscript) or Line 433-445 (in all-accepted manuscript).

2. The manuscript contains a number of grammatical and spelling mistakes that need to

be corrected. Below are some example. The authors should carefully edit the manuscript to address grammatical issues.

**Response 2:** Thanks for the helpful comments. We really appreciated the reviewer for carefully reviewing out manuscripts. We have carefully checked our manuscript and corrected faults.

Line 69: "in south hemisphere" should be "in southern hemisphere" Done

Line 85: "have been proposed could be considered" should be "have been proposed" Done

Line 115: "tidal-gravity wave during" should be "tidal-gravity wave interactions during" Done

Line 340: "Hence, the phase of (1,1) mode is focused." should be "Hence, we focus on the phase of the (1,1) mode" Done

Line 343: "We apply the method following" should be "We apply the following method" Done

Line 343: "devotion" should be "deviation" Done

Line 345 (and elsewhere): "confidential interval" should be "confidence interval" Done

Lines 390-391: "QBOW about 2 km" should be "QBOW by about 2 km) Done

Line 513: "thick in stratosphere" should be "thick in the stratosphere" Done

Line 516: "tides compare to QBOE" should be "tides compared to QBOE" Done

Line 556: "at subtropical" should be "at subtropical latitudes" Done